# Generalization of Gibbs and Langevin Monte Carlo Algorithms in the Interpolation Regime

**Andreas Maurer** [1]   **Erfan Mirzaei** [1 2 3]   **Massimiliano Pontil** [1 4]

## Abstract

This paper provides data-dependent bounds on the expected error of the Gibbs algorithm in the overparameterized interpolation regime, where low training errors are also obtained for impossible data, such as random labels in classification. The results show that generalization in the low-temperature regime is already signaled by small training errors in the noisier high-temperature regime. The bounds are stable under approximation with Langevin Monte Carlo algorithms. The analysis motivates the design of an algorithm to compute bounds, which on the MNIST, CIFAR-10 and SVHN datasets yield nontrivial, close predictions on the test error for true labeled data, while maintaining a correct upper bound on the test error for random labels.

## 1. Introduction

Modern learning algorithms can achieve very small training errors on arbitrary data if the underlying hypothesis space is large enough. For meaningful data, the chosen hypotheses also tend to have small test errors, a fortunate circumstance that has given great technological and economic thrust to deep learning. Unfortunately, the same algorithms also achieve very small training errors for data specifically designed to produce very large test errors, such as random labels in classification. In such a situation, which we will loosely refer to as the *interpolation regime*, the hypothesis space and the training error do not suffice to predict the test error. The key to generalization must be more deeply buried in the data. While not so disquieting to practitioners, this mystery has troubled theoreticians for many years (Zhang et al., 2021), and it seems safe to say that the underlying mechanisms still have not been completely understood.

We are far from solving this riddle in generality, but for sufficiently close approximations of the Gibbs posterior, we show how nontrivial bounds on the test error can be recovered from the training data. The Gibbs posterior assigns probabilities that decrease exponentially with the training error of the hypotheses. The exponential decay parameter $\beta$ can be interpreted as an inverse temperature in an analogy to statistical physics. The Gibbs measure is a sufficient idealization to have tractable theoretical properties, but it is also the limiting distribution of several concrete stochastic algorithms, here summarized as Langevin Monte Carlo (LMC), including Stochastic Gradient Langevin Dynamics (SGLD), (Gelfand & Mitter, 1991; Welling & Teh, 2011), a popular modern learning algorithm.

When $\beta$ is large, and the hypothesis space is rich, these algorithms can reproduce the dilemma described above by achieving very small training errors on data designed to have large test errors. Our paper addresses this regime of the Gibbs posterior and makes the following three contributions:

- We give high-probability data-dependent bounds on the true error, both for a hypothesis drawn from the Gibbs posterior and for the posterior mean, assuming that we can freely draw samples from it. These bounds hold for the entire range of temperatures.

- For time-homogeneous Markov processes, which converge to the Gibbs posterior, we derive bounds valid along the entire training trajectory, sharpening a recent result of (Harel et al., 2026) and extending it to the low-temperature regime.

- We show that these bounds are stable under approximations of the Gibbs posterior in relative entropy. Given enough computing resources, this yields bounds for LMC algorithms, based on known results for non-convex sampling.

- Existing convergence guarantees for LMC are insufficient for both a practical and rigorous computation of these bounds on real-world problems. A heuristic calibration step, based exclusively on the training

[1]Computational Statistics and Machine Learning, Istituto Italiano di Tecnologia, Genoa, Italy [2]Department of Mathematics, Università di Genova, Genova, Italy [3]CMAP, École Polytechnique, Institut Polytechnique de Paris, Palaiseau, France [4]Department of Computer Science, University College London, London, UK. Correspondence to: Erfan Mirzaei <erfunmirzaei@gmail.com>.

*Proceedings of the 43$^{rd}$ International Conference on Machine Learning*, Seoul, South Korea. PMLR 306, 2026. Copyright 2026 by the author(s).

data, leads to very close upper bounds on the test error for various neural networks trained with LMC on the MNIST, CIFAR-10, and SVHN datasets.

The idea underlying our bound is the following. The PAC-Bayesian theorem or its single draw variant (McAllester, 1999; Alquier, 2024; Rivasplata et al., 2020) bounds the generalization error roughly proportional to the logarithm of the posterior density or its posterior expectation (the relative entropy to the prior). The log-density of the Gibbs posterior at inverse temperature $\beta$ has an explicit expression in terms of an integral from 0 to $\beta$ of mean training errors, a fact which seems to have been overlooked in the PAC-Bayesian analysis of generalization (Lemma 3.1 below). Substitution of this integral in the PAC-Bayesian theorem then gives a bound on the generalization error.

As an illustrative example: if the loss $\ell$ has values in $[0, 1]$, and we happen to draw from the Gibbs posterior at $\beta$ a hypothesis $h$ with training error $\hat{L}(h, \mathbf{x}) = 0$, then we have the following bound on the expected (true) error of this hypothesis.

$$E_x [\ell (h, x)] \leq \frac{2 \left(A + \ln \left(2\sqrt{n}/\delta\right)\right)}{n}, \qquad (1)$$

where $A$ is the area in Figure 1, $n$ is the sample size and $\delta$ is the confidence parameter.

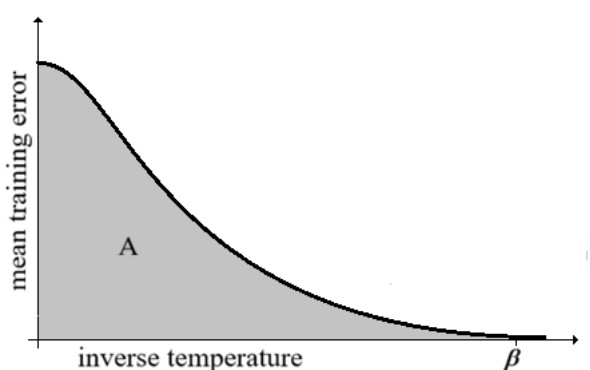

Figure 1. The mean training error of the Gibbs posterior is plotted against the inverse temperature. If the training error $\hat{L}(h, \mathbf{x})$ at $\beta$ is zero, then the log density, $\ln \frac{dG_\beta(\mathbf{x})}{d\pi}(h)$, is equal to the area $A$.

For the Gibbs posterior, this simple but novel reasoning resolves the dilemma of the interpolation regime. The training error of sufficiently overparametrized systems at a large value of $\beta$ (low temperature) is typically near zero and does not distinguish between easy and hard (e.g., random-label) data, but the mean training losses at small values of $\beta$ (high temperatures) will be quite different, leading to different predictions also at large $\beta$ (low temperature). Paraphrased:

*Better generalization in the low-temperature regime is al-*

*ready indicated by smaller training errors in the high-temperature regime.*

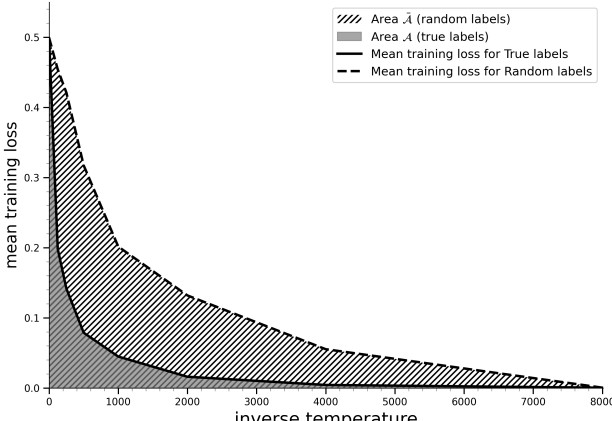

Figure 2. Mean training error of a fully-connected neural network with 400,000 parameters trained with SGLD on 2,000 examples of MNIST, both with true and random labels.

This principle appears to hold also for the distributions generated by practical LMC algorithms trained on real-world data, as is witnessed by Figure 2. In Section 5.4, we use the area ratio $A/\bar{A}$ and the fact that the true classification error for random labels is $1/2$, to develop a practical method to compute bounds in realistic environments.

After a brief survey of related literature, we review the PAC-Bayesian theorem, introduce the Gibbs posterior, and present our bounds, followed by an application to Markov processes and a stability analysis. We conclude with a section describing our experiments. The appendix contains a glossary of notation with section references.

## 1.1. Related literature

Many papers address the generalization of the Gibbs algorithm and Langevin Monte Carlo, with special focus on SGLD, which is the most popular algorithm. Most similar to this work is (Raginsky et al., 2017), which bounds the distance of the distribution generated by SGLD to the Gibbs posterior and then the latter's generalization error. Their bound applies only to the high-temperature regime $\beta < n$, but their convergence guarantees can be combined with our method to give bounds for SGLD in the entire temperature range.

Several works concentrate on the optimization path of SGLD. Mou et al. (2018) give both stability and PAC-Bayesian bounds. Pensia et al. (2018) apply the information-theoretic generalization bounds of (Xu & Raginsky, 2017). These ideas are further developed by Negrea et al. (2019), where random subsets of the training data are used to define data-dependent priors. Clerico et al. (2025) track the evolution of the posterior density along the trajectory of gra-

dient descent and use the single-draw version of PAC-Bayes (part (i) of Theorem 2.1 below) to obtain a bound in terms of norms of the Hessian summed over a fixed time horizon. Characteristically, the bound increases with training time and gives useful results only on a finite-time horizon. Farghly & Rebeschini (2021) give time-independent bounds for SGLD, which are further improved by Futami & Fujisawa (2024). Most of the bounds in the above papers are in expectation. The very recent paper of Harel et al. (2026) gives a very elegant argument for Markov chain algorithms based on the second law of thermodynamics. If the invariant distribution is the Gibbs posterior, the bound on the generalization gap along the entire optimization path is of order $\sqrt{\beta/n}$ but improvable to $\beta/n$.

Some papers give similar bounds for the Gibbs posterior, roughly of the form $\beta/n$ or $\sqrt{\beta/n}$ (Raginsky et al., 2017; Dziugaite & Roy, 2018; Kuzborskij et al., 2019; Rivasplata et al., 2020; Maurer, 2024; Harel et al., 2026)). These bounds hold equally for random labels and are therefore vacuous for overparametrized hypothesis spaces in the low temperature regime $\beta > n$. To our knowledge, ours is the only bound for the Gibbs posterior which is still informative in this regime.

The bound by Arora et al. (2019) is specialized to two-layer ReLU networks and derived from special properties of the gradient descent algorithm. Other bounds have been developed for specific algorithms designed to optimize them. The milestone paper by Dziugaite & Roy (2017) is the most prominent example; (Zhou et al., 2019), (Dziugaite & Roy, 2018) and (Pérez-Ortiz et al., 2021) are also in this category. Our bounds, by contrast, apply to the Gibbs posterior and LMC in their standard forms.

## 2. The PAC-Bayesian bound

Throughout the following $(\mathcal{X}, \Sigma)$ is a measurable space of *data* with probability measure $\mu$. The i.i.d. random vector $\mathbf{x} \sim \mu^n$ is the training sample.

$(\mathcal{H}, \Omega)$ is a measurable space of *hypotheses*, and $\ell : \mathcal{H} \times \mathcal{X} \to [0, \infty)$ is a prescribed loss function. Members of $\mathcal{H}$ are denoted $h$ or $g$. For a function $f : \mathcal{H} \to \mathbb{R}$, the sup-norm is denoted $||f||_\infty$. We write $L(h) := \mathbb{E}_{x \sim \mu}[\ell(h, x)]$ and $\hat{L}(h, \mathbf{x}) := (1/n)\sum_i \ell(h, x_i)$ respectively for the true (expected) and empirical error of hypothesis $h \in \mathcal{H}$.

The set of probability measures on $(\mathcal{H}, \Omega)$ is denoted $\mathcal{P}(\mathcal{H})$. The relative entropy or Kullback-Leibler-divergence between two probability measures is the function KL : $(\rho, \nu) \in \mathcal{P}(\mathcal{H}) \times \mathcal{P}(\mathcal{H}) \mapsto \mathbb{E}_{h \sim \rho}\left[\ln \frac{d\rho}{d\nu}(h)\right]$ if $\rho$ is absolutely continuous w.r.t. $\nu$, otherwise the value is $\infty$. The Rényi-infinity divergence (Rényi, 1961) is $R_\infty(\rho, \nu) = \sup_{h \in \mathcal{H}} \ln \frac{d\rho}{d\nu}(h)$ for $\nu, \rho \in \mathcal{P}(\mathcal{H})$. There is an a-priori reference measure $\pi \in \mathcal{P}(\mathcal{H})$, called the *prior*. A stochas-

tic algorithm is a function $\nu : \mathcal{X}^n \to \mathcal{P}(\mathcal{H})$, which assigns to a training sample $\mathbf{x}$ a probability measure $\nu(\mathbf{x}) \in \mathcal{P}(\mathcal{H})$.

The following general version of the PAC-Bayesian theorem appears in this form for the first time in (Rivasplata et al., 2020). It gives a bound for single hypotheses drawn from the posterior (i) as well as for posterior averages (ii). Appendix B.1 gives an easy proof for the reader's benefit.

**Theorem 2.1.** *Let* $F : \mathcal{H} \times \mathcal{X}^n \to \mathbb{R}$ *be some measurable function, and let* $\nu$ *be a stochastic algorithm such that* $\nu(\mathbf{x})$ *is absolutely continuous w.r.t.* $\pi$ *for all* $\mathbf{x} \in \mathcal{X}^n$. *Then*

*(i) for* $\delta > 0$ *with probability at least* $1 - \delta$ *in* $\mathbf{x} \sim \mu^n$ *and* $h \sim \nu(\mathbf{x})$

$$F(h, \mathbf{x}) \leq \ln \frac{d\nu(\mathbf{x})}{d\pi}(h) + \ln \frac{\mathbb{E}_{\mathbf{x}}\mathbb{E}_{g \sim \pi}\left[e^{F(g, \mathbf{x})}\right]}{\delta}$$

*(ii) for* $\delta > 0$ *with probability at least* $1 - \delta$ *in* $\mathbf{x} \sim \mu^n$

$$\mathbb{E}_{h \sim \nu(\mathbf{x})}[F(h, \mathbf{x})] \leq KL(\nu(\mathbf{x}), \pi) + \ln \frac{\mathbb{E}_{\mathbf{x}}\mathbb{E}_{g \sim \pi}\left[e^{F(g, \mathbf{x})}\right]}{\delta}$$

Here, $F$ is a placeholder for a random variable related to the generalization gap, which we want to bound. Suppose $\ell$ has values in $[0, 1]$. With a suitable choice of $F$, we can use part (i) above to derive, with probability at least $1 - \delta$ as $\mathbf{x} \sim \mu^n$ and $h \sim \nu(\mathbf{x})$, that

$$
\begin{aligned}
L(h) \leq{} & \hat{L}(h, \mathbf{x}) + \sqrt{\frac{2\hat{L}(h, \mathbf{x})}{n}\left(\ln \frac{d\nu(\mathbf{x})}{d\pi} + \ln \frac{2\sqrt{n}}{\delta}\right)} \\
& + \frac{2}{n}\left(\ln \frac{d\nu(\mathbf{x})}{d\pi} + \ln \frac{2\sqrt{n}}{\delta}\right).
\end{aligned}
\tag{2}
$$

Note that for $\hat{L}(h, \mathbf{x}) = 0$ and $\ln \frac{d\nu(\mathbf{x})}{d\pi}(h) = A$ we obtain the bound in Equation (1). From (ii) we get the analogous bound, if $L(h)$ is replaced by $\mathbb{E}_{h \sim \nu(\mathbf{x})}[L(h)]$, $\hat{L}(h, \mathbf{x})$ by $\mathbb{E}_{h \sim \nu(\mathbf{x})}\left[\hat{L}(h, \mathbf{x})\right]$ and $\ln \frac{d\nu(\mathbf{x})}{d\pi}$ by $KL(\nu(\mathbf{x}), \pi)$. Details are in Appendix B. For more information on PAC-Bayesian theory, we refer to the treatises of (Guedj, 2019; Alquier, 2024).

Clearly, the crucial terms in these bounds are $\ln \frac{d\nu(\mathbf{x})}{d\pi}(h)$ or $KL(\nu(\mathbf{x}), \pi)$ respectively. In the sequel, we concentrate on bounding these quantities and often omit their mechanical resubstitution in Theorem 2.1.

Several variants of the PAC-Bayesian theorem are minimized by the Gibbs algorithm, which we introduce in the next section (see (McAllester, 1999; Guedj, 2019; Alquier, 2024)). It is therefore a natural candidate to study the power and the limitations of PAC-Bayesian theory.

## 3. Bounds for the Gibbs algorithm

With a fixed prior, the Gibbs algorithm at inverse temperature $\beta > 0$ is the stochastic algorithm $G_\beta : \mathbf{x} \in \mathcal{X}^n \mapsto$

$G_\beta(\mathbf{x}) \in \mathcal{P}(\mathcal{H})$ defined by

$$G_\beta(\mathbf{x})(A) = \frac{1}{Z_\beta(\mathbf{x})} \int_A e^{-\beta \hat{L}(h,\mathbf{x})} d\pi(h) \text{ for } A \in \Omega.$$

$G_\beta(\mathbf{x})$ is called the *Gibbs posterior,* the normalizing factor

$$Z_\beta(\mathbf{x}) := \int_{\mathcal{H}} e^{-\beta \hat{L}(h,\mathbf{x})} d\pi(h)$$

is called the *partition function*.

The Gibbs posterior provides a principled, albeit idealized, way to put larger weights on hypotheses with smaller empirical error. As $\beta \to \infty$, the Gibbs posterior concentrates on the set of empirical risk minimizers (Athreya & Hwang, 2010), so the low-temperature regime (equivalent to large $\beta$) is particularly interesting.

Evidently $\ln(dG_\beta(\mathbf{x})/d\pi)(h) = -\beta \hat{L}(h,\mathbf{x}) - \ln Z_\beta(\mathbf{x})$. This function has an important integral representation, which is well known from statistical mechanics (see, e.g. (Huang, 2008)). Despite its simplicity, it seems to have been overlooked in the literature on generalization. We found the work by Ujváry et al. (2023) that uses parts of the following lemma for a similar idea, only after submitting this work.

**Lemma 3.1.** *For all $\beta \geq 0$, $\mathbf{x} \in \mathcal{X}^n$ and $h \in \mathcal{H}$*

$$-\ln Z_\beta(\mathbf{x}) = \int_0^\beta \mathbb{E}_{g \sim G_\gamma(\mathbf{x})}\left[\hat{L}(g,\mathbf{x})\right] d\gamma \quad (3)$$

$$\ln \frac{dG_\beta(\mathbf{x})}{d\pi}(h) = \int_0^\beta \left( \mathbb{E}_{g \sim G_\gamma(\mathbf{x})}\left[\hat{L}(g,\mathbf{x})\right] \right. \\ \left. - \hat{L}(h,\mathbf{x}) \right) d\gamma \quad (4)$$

$$KL(G_\beta(\mathbf{x}), \pi) = \int_0^\beta \left( \mathbb{E}_{g \sim G_\gamma(\mathbf{x})}\left[\hat{L}(g,\mathbf{x})\right] \right. \\ \left. - \mathbb{E}_{h \sim G_\beta(\mathbf{x})}\left[\hat{L}(h,\mathbf{x})\right] \right) d\gamma. \quad (5)$$

*Also the function $\beta \mapsto \mathbb{E}_{g \sim G_\beta(\mathbf{x})}\left[\hat{L}(g,\mathbf{x})\right]$ is non-increasing in $\beta$.*

*Proof.* Let $A(\beta) = -\ln Z_\beta(\mathbf{x})$. One verifies the identities

$$A(0) = 0,$$
$$A'(\beta) = \frac{1}{Z_{\beta,\pi}(\mathbf{x})} \int_{\mathcal{H}} \hat{L}(h,\mathbf{x}) e^{-\beta \hat{L}(h,\mathbf{x})} d\pi(h)$$
$$= \mathbb{E}_{h \sim G_\beta(\mathbf{x})}\left[\hat{L}(h,\mathbf{x})\right],$$
$$A''(\beta) = -\text{Var}_{h \sim G_\beta(\mathbf{x})}\left[\hat{L}(h,\mathbf{x})\right] \leq 0,$$

where Var denotes variance. Equation (3) then follows from the first two of these identities and the fundamental theorem

of calculus, and the last assertion in the Lemma follows from the last identity. Since the logarithm of the density of the Gibbs posterior is $-\beta \hat{L}(.,\mathbf{x}) - \ln Z_\beta(\mathbf{x})$ we get Eq. (4). Then Eq. (5) follows from taking the expectation of Eq. (4) in $G_\beta(\mathbf{x})$. □

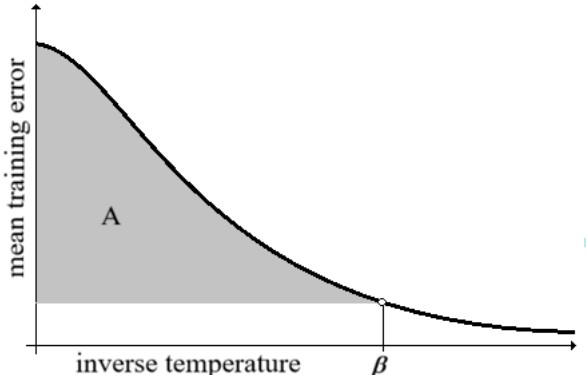

*Figure 3.* The mean training error of the Gibbs posterior is plotted against the inverse temperature. The relative entropy $KL(G_\beta(\mathbf{x}), \pi)$ is equal to the shaded area $A$.

Figure 3 provides a simple geometrical interpretation for Eq. (5). For Eq. (4), if the shaded area were equal to $\ln(dG_\beta(\mathbf{x})/d\pi)(h)$, its lower boundary would fluctuate with a variance equal to the negative slope at $\beta$, depending on the draw of $h$ (compare the proof of Lemma 3.1).

The identities (4) and (5) of Lemma 3.1 can now be substituted into the PAC-Bayesian Theorem (Theorem 2.1) to give our bounds for the Gibbs algorithm. That they are completely determined by the mean training errors at higher temperatures (smaller $\beta$) is evident from Figure 3. If these errors decrease quickly, we expect better generalization; if they decrease slowly, we expect worse generalization.

## 4. Bounds for Markov processes and Langevin Monte Carlo

The Gibbs posterior is an idealization, from which it is impossible to sample directly. Nevertheless, several works describe Markov processes, here summarized as Langevin Monte Carlo (LMC), capable of approximating a probability measure $\nu$ on $\mathbb{R}^d$ of the form $\nu \propto \exp(-V)$, or some nearby limiting distribution.

The classical prototype is Continuous Langevin Dynamics (CLD), a Markov process in continuous time describing thermalization in statistical physics and originating in the study of Brownian motion (Langevin et al., 1908). To turn the continuous process into an iterative algorithm, several discretized versions have been proposed. There is the Metropolis Adjusted Langevin Algorithm (MALA) (Roberts & Tweedie, 1996), which is the Euler-discretization of CLD

with an additional Metropolis-style accept-reject step to ensure that the invariant distribution is indeed the desired Gibbs distribution. Hamiltonian Monte Carlo Markov Chain (HMCMC) (Duane et al., 1987; Betancourt, 2017) is a refinement of MALA. The Unadjusted Langevin Algorithm (ULA) (Parisi, 1981) is the discretization of CLD without the accept-reject step and converges to a slightly different distribution. Stochastic Gradient Langevin Dynamics (SGLD) (Gelfand & Mitter, 1991; Welling & Teh, 2011) is an accelerated version of ULA replacing the true gradient of $V$ by an unbiased estimate realized with minibatches. All these processes are of theoretical interest as models for Stochastic Gradient Descent (SGD), but at least SGLD is also used as a learning algorithm in practice. The method of parallel tempering (Syed et al., 2022) seems promising for the computation of our bounds. In Appendix C.1, we give more recent references containing convergence guarantees and discuss CLD and ULA in some detail.

In the next section we show how the PAC-Bayesian bound and the integral representation can be applied to time-homogeneous Markov processes.

Throughout this section, we assume $\mathcal{H} = \mathbb{R}^d$ and an isotropic Gaussian prior $\pi$ of width $\sigma$. We condition on the training data $\mathbf{x}$, reference to which we often omit.

## 4.1. Bounds for Markov processes

Here, we both apply and sharpen the method in the recent work of Harel et al. (2026) on Markov processes. We take $\{h_t(\mathbf{x})\}_{t \in I}$ to be a time-homogeneous Markov process in real or discrete time, $I = [0, \infty)$ or $I = \mathbb{N}_0$, with values in $\mathcal{H}$. This is our model of a training process, such as CLD or the practically implementable MALA, ULA, or SGLD. The distribution of $h_t(\mathbf{x})$ will be denoted $\nu_t(\mathbf{x})$. We assume that these and possible invariant distributions have positive densities with respect to $\pi$. This assumption captures the relevant practical cases under consideration. The following lemma is sometimes referred to as the *second law of thermodynamics*. A proof is given in Section C.3.

**Lemma 4.1.** *(Second law of thermodynamics) If $\nu$ is a stationary distribution of $\{h_t\}_{t \in I}$ and $s < t$ then $KL(\nu_t, \nu) \leq KL(\nu_s, \nu)$ and $R_\infty(\nu_t, \nu) \leq R_\infty(\nu_s, \nu)$, with equality in either case if and only if $\nu_s = \nu$.*

Now for any $t \in I$ and any $\beta > 0$

$$KL(\nu_t, \pi) = \mathbb{E}_{h \sim \nu_t}\left[\ln \frac{d\nu_t}{dG_\beta}\right] + \mathbb{E}_{h \sim \nu_t}\left[\ln \frac{dG_\beta}{d\pi}\right]$$
$$= KL(\nu_t, G_\beta) - \beta \mathbb{E}_{h \sim \nu_t}\left[\hat{L}(h)\right] - \ln Z_\beta. \quad (6)$$

There is an analogous identity for the single draw, omitting the expectations in $\nu_t$ and replacing $KL$ by $R_\infty$. Now assume that $G_\beta$ is a stationary distribution of the process, and that $\nu_0 = \pi$, so the process is started from the

prior. Then Harel et al. (2026) use Lemma 4.1 above, to get $KL(\nu_t, G_\beta) \leq KL(\pi, G_\beta) = \beta \mathbb{E}_{h \sim \pi}\left[\hat{L}(h)\right] + \ln Z_\beta$, and they substitute this bound in the above identity. Since the partition functions cancel each other, and since $\beta \mathbb{E}_{h \sim \nu_t}\left[\hat{L}(h)\right] \geq 0$, they end up with $KL(\nu_t, \pi) \leq \beta \mathbb{E}_{h \sim \pi}\left[\hat{L}(h)\right]$, to be substituted in the PAC-Bayesian bound. There is a similar bound for the single draw in terms of $R_\infty$. The resulting generalization bounds are valid along the *entire training trajectory*. They are, however, largely independent of distribution and data, and vacuous for $\beta > n$, since typically $\mathbb{E}_{h \sim \pi}\left[\hat{L}(h)\right]$ and $\left\|\hat{L}\right\|_\infty$ are on the order of unity or larger (here, we omitted several substantial refinements in (Harel et al., 2026)).

The use of the second law is elegant, but in this form, it forgoes the potential benefit of a process converging to $G_\beta$, such as CLD, MALA, or HMCMC, or to some nearby distribution such as ULA or SGLD. The following proposition takes advantage of convergence as well as of close approximation.

**Proposition 4.2.** *Assume $\nu_0 = \pi$.*
*(i) Let $\lambda_t = R_\infty(\nu_t, G_\beta) / R_\infty(\pi, G_\beta)$. Then*

$$R_\infty(\nu_t, \pi) \leq \lambda_t \beta \left\|\hat{L}\right\|_\infty + (1 - \lambda_t) \int_0^\beta \mathbb{E}_{g \sim G_\gamma}\left[\hat{L}(g)\right] d\gamma.$$

*(ii) If instead $\lambda_t = KL(\nu_t, G_\beta) / KL(\pi, G_\beta)$, then*

$$KL(\nu_t, \pi) \leq \lambda_t \beta \mathbb{E}_{g \sim \pi}[\hat{L}(g)] + (1 - \lambda_t) \int_0^\beta \mathbb{E}_{g \sim G_\gamma}[\hat{L}(g)] d\gamma.$$

*Proof.* We only prove (i), the proof of (ii) being analogous.

$$
\begin{aligned}
R_\infty(\nu_t, \pi) &= \sup_{h \in \mathcal{H}}\left(\ln \frac{d\nu_t}{dG_\beta}(h) + \ln \frac{dG_\beta}{d\pi}(h)\right) \\
&\leq \lambda_t R_\infty(\pi, G_\beta) + \sup_{h \in \mathcal{H}}\left(-\beta \hat{L}(h) - \ln Z_\beta\right) \\
&\leq \lambda_t \sup_{h \in \mathcal{H}}\left(\beta \hat{L}(h) + \ln Z_\beta\right) - \ln Z_\beta \\
&= \lambda_t \beta \left\|\hat{L}\right\|_\infty + (1 - \lambda_t) \int_0^\beta \mathbb{E}_{h \sim G_\gamma}[\hat{L}(h)] d\gamma,
\end{aligned}
$$

where we used Lemma 3.1 in the last step. $\square$

**Remarks:** 1. Without any additional assumptions, these bounds hold for all $\beta$ and along the entire training trajectory. By the last assertion of Lemma 3.1 the integral is generically smaller than $\beta \left\|\hat{L}\right\|_\infty$ or $\beta \mathbb{E}_{h \sim \pi}\left[\hat{L}(h)\right]$ respectively. Thus, whenever $\lambda_t < 1$, the bounds are strictly smaller than those of (Harel et al., 2026).

2. If $G_\beta$ is invariant, then by the second law (Lemma 4.1) $\lambda_t$ is strictly decreasing in $t$ and $\lambda_t < 1$ for $t > 0$ in all non-trivial cases. The bounds then move in convex interpolation towards the integral.

3. If the process converges to $G_\beta$ in relative entropy, meaning that $KL\left(\nu_t, G_\beta\right) \to 0$, then $\lambda_t \to 0$ as $t \to \infty$. In this case, the bounds asymptotically approach those of the Gibbs posterior in Lemma 3.1, with the modification that the negative terms are omitted. For large $t$, they exhibit the same sensitivity of generalization to mean training errors of the Gibbs posterior at higher temperatures. This is the case for CLD and all algorithms containing a Metropolis-style accept-reject mechanism, such as MALA or HMCMC.

To illustrate this, we adapt the stochastic differential equation for CLD to temperature and prior (see Section C.1). It becomes

$$dh_{\beta,t} = -\left(\nabla \hat{L}\left(h_{\beta,t}\right) + \frac{h_{\beta,t}}{\beta\sigma^2}\right)dt + \sqrt{\frac{2}{\beta}}dB_t,$$

where $B_t$ is standard centered Brownian motion in $\mathbb{R}^d$. Let $\nu_{\beta,t}$ be the distribution of $h_{\beta,t}$ at time $t$. In Lemma C.3 we show that, if $G_\beta$ satisfies a logarithmic Sobolev inequality with constant $\alpha$ (see Section C.1), then $KL\left(\nu_{\beta,t}, G_\beta\right) \leq e^{-2\alpha t/\beta} KL\left(\nu_{\beta,0}, G_\beta\right)$. So if CLD starts from the prior, we can use Proposition 4.2 (ii) with $\lambda_t = e^{-2\alpha t/\beta}$ and obtain

$$
\begin{aligned}
KL\left(\nu_{\beta,t}, \pi\right) \quad \leq \quad & e^{-2\alpha t/\beta}\, \beta\, \mathbb{E}_{h\sim\pi}\left[\hat{L}\left(h\right)\right] \\
& + (1 - e^{-2\alpha t/\beta})\int_0^\beta \mathbb{E}_{G_\gamma}\left[\hat{L}\left(g\right)\right]d\gamma.
\end{aligned}
$$

### 4.2. Stability of the Bounds

If $\nu : \mathcal{X}^n \to \mathcal{P}(\mathcal{H})$ is the stochastic algorithm for which we want the bound, then Eq. (6) and the results of the previous section show that with sufficient approximation of $\nu\left(\mathbf{x}\right)$ by $G_\beta\left(\mathbf{x}\right)$ in relative entropy, we can largely recover the bounds for the Gibbs posterior. But these bounds, though data-dependent, are still inaccessible because of the continuous nature of the temperature integral and the impossibility to sample directly from the Gibbs posterior. We now study the following question of stability: given some algorithm to approximate $G_\beta$ for any $\beta$ to arbitrary precision in relative entropy, can we also approximate our bounds to arbitrary precision?

To this end, we discretize the temperature scale of the integral and approximate each $G_{\beta_k}$ by some distribution $\nu_k$. The error incurred on the corresponding expectations of $\hat{L}$ can then be controlled under either boundedness or Lipschitz conditions on the loss $\ell$.

**Definition 4.3.** For $\mathbf{x} \in \mathcal{X}^n$, an increasing sequence $\beta_0^K = (0 = \beta_0 < \beta_1 < \cdots < \beta_K = \beta)$ of positive numbers, and

a corresponding vector of data-dependent distributions $\nu_0^{K-1}(\mathbf{x}) = \left(\nu_0\left(\mathbf{x}\right), \nu_1\left(\mathbf{x}\right), \cdots, \nu_{K-1}\left(\mathbf{x}\right)\right) \in \mathcal{P}\left(\mathcal{H}\right)^K$ we denote

$$\Gamma(\nu_0^{K-1}, \mathbf{x}, \beta_0^K) = \sum_{k=1}^K (\beta_k - \beta_{k-1})\mathbb{E}_{g\sim\nu_{k-1}(\mathbf{x})}\left[\hat{L}(g, \mathbf{x})\right].$$

For an illustration see Figure 5 in Section C.4. The next lemma bounds the estimation error relative to the temperature integral in terms of the relative entropies.

**Lemma 4.4.** *With $\beta_0^K$ and $\nu_0^{K-1}$ as in Definition 4.3 denote*

$$\Delta := \int_0^\beta \mathbb{E}_{h\sim G_\gamma(\mathbf{x})}\left[\hat{L}\left(h, \mathbf{x}\right)\right]d\gamma - \Gamma(\nu_0^{K-1}, \mathbf{x}, \beta_0^K)$$

*(i) If $\mathbb{E}_{h\sim G_{\beta_k}(\mathbf{x})}\left[\hat{L}\left(h, \mathbf{x}\right)\right] \leq \mathbb{E}_{h\sim\nu_k(\mathbf{x})}\left[\hat{L}\left(h, \mathbf{x}\right)\right]$ for all $k$ and $\mathbf{x}$, then $\Delta \leq 0$.*

*(ii) If $\ell\left(h, \mathbf{x}\right)$ is bounded in $h$ for all $\mathbf{x}$, $\|\ell\left(h, \mathbf{x}\right)\| \leq m$ then*

$$\Delta \leq m\sum_{k=1}^K (\beta_k - \beta_{k-1})\sqrt{KL\left(\nu_{k-1}\left(\mathbf{x}\right), G_{\beta_{k-1}}\left(\mathbf{x}\right)\right)/2}.$$

*(iii) If instead $\ell\left(h, \mathbf{x}\right)$ is $m$-Lipschitz in $h$ for all $\mathbf{x}$, $\ell\left(h, \mathbf{x}\right) - \ell\left(g, \mathbf{x}\right) \leq m\left\|h - g\right\|$ and $G_{\beta_k}\left(\mathbf{x}\right)$ satisfies an LSI with constant $\alpha$ for all $k$ and $\mathbf{x}$, then*

$$\Delta \leq \frac{2m}{\alpha}\sum_{k=1}^K (\beta_k - \beta_{k-1})KL\left(\nu_{k-1}\left(\mathbf{x}\right), G_{\beta_{k-1}}\left(\mathbf{x}\right)\right).$$

By the last conclusion of Lemma 3.1, part (i) is immediate. Proofs of parts (ii) and (iii) are given in Appendix C.4. The assumption in case (i) is not implausible if we start LMC from a non-informative prior, and in our experiments, we always observed decreasing losses along the LMC path. The analysis sketched in Section C.7 justifies this assumption for ULA in a simplified linear scenario.

The next theorem gives our final bound in terms of arbitrary distributions and their relative entropies to Gibbs distributions.

**Theorem 4.5.** *Let $F : \mathcal{H} \times \mathcal{X}^n \to \mathbb{R}$ be some measurable function and $\beta_0^K$ and $\nu_0^{K-1}$ as in Definition 4.3. Let $\nu(\mathbf{x})$ be any data-dependent distribution on $\mathcal{H}$. Let $\Delta$ be bounded as in Lemma 4.4, depending on which of the conditions is fulfilled by $\ell$. Then*

*(i) with probability at least $1 - \delta$ as $\mathbf{x} \sim \mu^n$ and $h \sim \nu(\mathbf{x})$*

$$
\begin{aligned}
F\left(h, \mathbf{x}\right) \quad \leq \quad & -\beta\hat{L}\left(h, \mathbf{x}\right) + \Gamma(\nu_0^{K-1}, \mathbf{x}, \beta_0^K) \\
& + \ln\mathbb{E}_{\mathbf{x}}\mathbb{E}_{h\sim\pi}\left[e^{F(h,\mathbf{x})}\right] + \ln\frac{1}{\delta} \\
& + R_\infty\left(\nu\left(\mathbf{x}\right), G_\beta\left(\mathbf{x}\right)\right) + \Delta.
\end{aligned}
$$

*If $F$ and $\ell$ are bounded, then $R_\infty\left(\nu\left(\mathbf{x}\right), G_\beta\left(\mathbf{x}\right)\right)$ can be replaced by*

$$\max\left\{0, \beta\left\|\ell\right\|_\infty + \left\|F\right\|_\infty + \ln\sqrt{2KL\left(\nu\left(\mathbf{x}\right), G_\beta\left(\mathbf{x}\right)\right)}\right\}.$$

*(ii) with probability at least $1 - \delta$ as $\mathbf{x} \sim \mu^n$*

$$
\begin{aligned}
\mathbb{E}_{\nu(\mathbf{x})}\left[F(h, \mathbf{x})\right] \leq\ & -\beta\,\mathbb{E}_{\nu(\mathbf{x})}\left[\hat{L}\left(h, \mathbf{x}\right)\right] + \Gamma(\nu_0^{K-1}, \mathbf{x}, \beta_0^K) \\
& + \ln\mathbb{E}_\mathbf{x}\mathbb{E}_{h\sim\pi}\left[e^{F(h,\mathbf{x})}\right] + \ln\frac{1}{\delta} \\
& + KL\left(\nu\left(\mathbf{x}\right), G_\beta\left(\mathbf{x}\right)\right) + \Delta.
\end{aligned}
$$

The left-hand side in both inequalities is the random variable that we wish to bound, depending on the distribution $\nu$. The right-hand side of the first two lines is the bound, as computed from the distributions $\nu(\mathbf{x})$ and $\nu_0^{K-1}(\mathbf{x})$ and includes the dependence on the exponential moment of $F$ and the confidence parameter $\delta$. The third line gives the error incurred by the fact that none of the distributions is really the right Gibbs distribution. The first term there gives the error for the target distribution $\nu(\mathbf{x})$, and is different in the single-draw and classical PAC-Bayesian cases. The term $\Delta$ results from approximating the temperature integral by the expectations in a finite number of distributions, as described in Lemma 4.4.

The amendment to (i) is necessary, since we know of no process with useful bounds on $R_\infty$. The replacement indeed converges to zero with relative entropy, but, since $\beta\left\|\ell\right\|_\infty + \left\|F\right\|_\infty$ is typically of order $n$, it requires the relative entropy to be exponentially small in $n$. Nevertheless, if the bounds in Corollary C.2 are substituted in part (ii) or in the amendment to part (i), they guarantee, with sufficient computational budget and appropriate choices of $t$ and $\eta$, the convergence of LMC to our bounds for the Gibbs posterior.

A detailed proof of Theorem 4.5 is given in Appendix C.4. Parts (ii) and (i) without amendment follow more or less directly from the PAC-Bayesian theorem, Lemma 4.4, and equation (6) and a reasoning using $R_\infty$ analogous to (6). The amendment to (i) requires a special adapted proof of the PAC-Bayesian theorem.

# 5. Experiments

The purpose of our experiments is twofold. For one, we want to verify that the theoretical dependence of the generalization performance of the Gibbs posterior at low temperatures on its training errors at high temperatures carries over to practical temperature-regularized algorithms like SGLD in real-world settings, with overparametrized neural networks. Indeed, in all our experiments, the temperature plots of the mean training errors, computed as described below, verify the qualitative prediction that both the failure of generalization for random labels and the success for true labels are related to the areas under the curves at higher temperatures (see, for example, Figure 2).

Second, we would like to use the training data, and only the training data, to make realistic quantitative predictions of test errors in such settings. This is more difficult, since the guarantees of Corollary C.2 in combination with Theorem 4.5 are inadequate in high-dimensional situations. We achieve this goal with a principled calibration scheme described below.

## 5.1. Experimental Environment and Algorithms

The real-world data are either the MNIST dataset, subdivided into the two classes of characters 0–4 and 5–9, or the CIFAR-10 dataset to distinguish between animals and vehicles. For impossible data, we randomize the labels of the training data. Our experiments are computationally heavy, so we generally use small sample sizes, from 2000 to 8000 examples. For comparison to the baselines, we used the whole MNIST dataset samples for a multi-class classification task to compare our results with the empirical baseline. The hypothesis space is the set of weight vectors for a neural network with ReLU activation functions constrained by a Gaussian prior distribution with $\sigma = 5$. Neural network architectures are described in Section D.1.1 of the appendix. To approximately sample the weight vectors in the vicinity of the Gibbs posterior, we use ULA as in (16) or SGLD (Welling & Teh, 2011) with constant step size $\eta$.

## 5.2. The Loss Function $\ell$

Most experiments were done with bounded loss functions $\ell$, either bounded binary cross-entropy as described in Appendix D of (Dziugaite & Roy, 2018) or the Savage loss (Masnadi-Shirazi & Vasconcelos, 2008). As an unbounded loss function, we tried binary cross-entropy (BCE) (Section D.2.6), but with a smaller value of $\sigma$, so as to avoid excessive training errors for small values of $\beta$. We compute bounds for the 0-1 loss, using the method described in Section B.2.

## 5.3. Approximating the Ergodic Mean

As we know of no sufficient criterion for convergence, we terminate iterations at time $T$, when a very slow running mean $\mathbb{M}_{\text{stop}}$ of the loss trajectory $\left(\hat{L}(h_{\beta_k, t}, \mathbf{x})\right)_{t=0}^T$ stops decreasing. A second running mean $\mathbb{M}_{\text{erg}}$ is used as an approximation of the ergodic mean and thus of expectations in the invariant distribution. We thus replace all expectations $\mathbb{E}_{h\sim G_{\beta_k}}\left[\hat{L}(h, \mathbf{x})\right]$ occurring in the bounds by $\mathbb{M}_{\text{erg}}\left[\left(\hat{L}(h_{\beta_k, t}, \mathbf{x})\right)_{t=0}^T\right]$. Both running means $\mathbb{M}_{\text{stop}}$ and $\mathbb{M}_{\text{erg}}$ are implemented as first-order, recursive lowpass filters described in Section D.1.3 of the appendix.

## 5.4. Computation of the Bounds and Calibration

For the 01-error (cf. Section B.2) we compute our bounds from the PAC-Bayesian theorem in the form

$$\mathbb{E}_{h \sim \nu_\beta(\mathbf{x})} \left[ L_{01}(h) \right] \tag{7}$$

$$\leq \kappa^{-1} \left( \mathbb{E}_{h \sim \nu_\beta(\mathbf{x})} \left[ \hat{L}_{01}(h, \mathbf{x}) \right], \frac{\mathcal{Q} + \ln\left( \frac{2\sqrt{n}}{\delta} \right)}{n} \right),$$

where $\nu_\beta(\mathbf{x})$ is an LMC approximation to $G_\beta(\mathbf{x})$ and $\mathcal{Q}$ is a proxy for $KL(\nu_\beta(\mathbf{x}), \pi)$, the computation of which is described below. Using $\kappa^{-1}$ is somewhat more accurate than the analog of (2). The function $\kappa$ is the relative entropy of two Bernoulli variables, the derivation of the above bound and the inverse function $\kappa^{-1}$ are explained in section B.2.

Our approximation to $KL(\nu_\beta(\mathbf{x}), \pi)$ is

$$A = -\beta \, \mathbb{E}_{h \sim \nu_\beta} \left[ \hat{L}(h, \mathbf{x}) \right] + \Gamma(\nu_0^{K-1}, \mathbf{x}, \beta_0^K).$$

For a rigorous bound following Theorem 4.5 we would have to set $\mathcal{Q} = A$ plus all the terms bounding the approximation errors as in Corollary C.2. Unfortunately, the quantities $R$ and $\alpha$ are impossible to estimate in practice. But even if we assume these to be in the order of unity, the bounds are too coarse to distinguish between different temperatures with realistic step sizes. A simple calculation (see Lemma C.7 in Appendix C.5) shows that

$$KL(G_\beta, G_{2\beta}) \leq \beta \left( \mathbb{E}_{G_\beta} \left[ \hat{L} \right] - \mathbb{E}_{G_{2\beta}} \left[ \hat{L} \right] \right) \leq \beta \mathbb{E}_{G_\beta} \left[ \hat{L} \right].$$

By Corollary C.2 we should therefore have at least $8\eta d R^2 / \alpha < \mathbb{E}_{G_\beta(\mathbf{x})} \left[ \hat{L} \right]$ to distinguish between the expectations in the Gibbs posterior for $\beta$ and $2\beta$. The smallest neural network we use has $d = 392,500$. If $\ell$ has values in $[0, 1]$ then $\mathbb{E}_{G_\beta(\mathbf{x})} \left[ \hat{L} \right] \leq 1$, so even if $R$ and $\alpha$ are set to 1, we would need step sizes in the order of $10^{-7}$. Safe values of $\eta$, as suggested by the theoretical results in Section 4, are therefore impossible in practice, and the bound has to be adapted to a realistic choice of $\eta$.

But the data for the true labels $\mathbf{x}$ and the data $\bar{\mathbf{x}}$ for the random labels are strongly related. Dimension and input marginals are equal. This suggests that the inaccuracy of the LMC approximations affects both in a similar way, and we make the following calibration assumption:

$$\frac{KL(\nu_\beta(\mathbf{x}), \pi)}{KL(\nu_\beta(\bar{\mathbf{x}}), \pi)} = \frac{A}{\bar{A}}, \tag{8}$$

where $\bar{A}$ is given by the expression analogous to $A$ for the random labels, compare Figure 2.

The expected error of the random labels in binary classification is $1/2$. Now let $r$ be the smallest positive factor

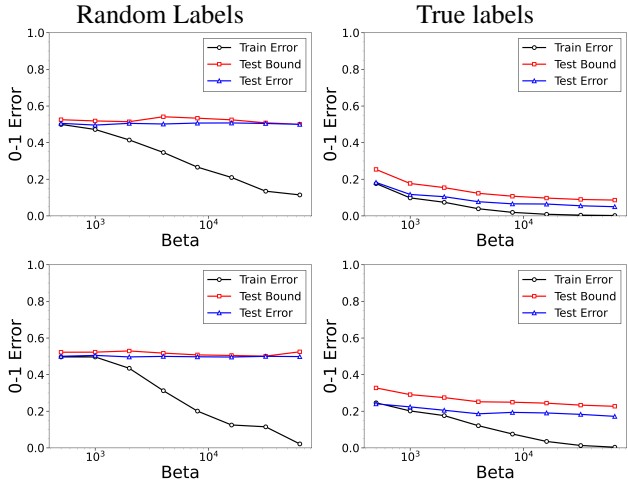

*Figure 4.* SGLD on MNIST and CIFAR-10 with 8000 training examples, MNIST above and CIFAR-10 below, random labels on the left, true labels on the right. Both random and true labels are trained with the same algorithm and parameters on a fully connected ReLU network with two hidden layers of 1000 and 1500 units, respectively. The calibration factor for MNIST is 0.77, for CIFAR-10, 0.89. Train error, test error, and our bound for the Gibbs posterior average of the 0-1 loss are plotted against $\beta$.

such that $r\bar{A}$, when substituted for $\mathcal{Q}$ in (7) yields a value greater or equal $1/2$. If the PAC-Bayesian theorem is tight, then this means that $r\bar{A} \geq KL(\nu_\beta(\bar{\mathbf{x}}), \pi)$. Our calibration assumption above then also implies that

$$rA \geq KL(\nu_\beta(\mathbf{x}), \pi),$$

and substitution of $rA$ for $\mathcal{Q}$ in (7) should result in a correct upper bound for the true labels. For a precise definition of $r$ see Section C.6.

In Section C.7 we sketch an argument that Eq. 8 holds as $\beta$ goes to infinity for Bayesian linear regression with quadratic loss under the assumption of a Gaussian prior. The main idea is that we can decompose the KL term and the area into label-dependent and label-independent parts. The label-dependent term remains constant, while the label-independent term grows logarithmically. Therefore, the ratio holds in the low-temperature limit.

In addition to having a justification for this simplified setting, we empirically found that the choice of $r$ leads to correct and surprisingly tight upper bounds on the test error of correctly labeled data in all cases we tried. We emphasize that our calibration procedure depends only on the training data.

## 5.5. Results

Several experiments confirm the validity of the proposed bounds. An example is shown in Figure 4, where a fully connected ReLU network with two hidden layers of 1000 (respectively 1500) units each is trained with SGLD at inverse temperatures $\beta = 0, 500, 1000, 2000, 4000, 8000,$

16000, 32000, and 64000. The train error for random labels is about 0.1 (or even less) at $\beta = 64000$, where the bound is above 0.5. The test error for true labels, however, is tightly bounded above.

Notice that for MNIST, which has the tightest bounds, the training error for the true labels is rapidly decreasing from 0.5 to 0.17 at $\beta = 500$ and to 0.1 at $\beta = 1000$. The more moderate initial decrease for CIFAR-10 corresponds to the tendency to overfit on this more difficult dataset. This confirms the intuition that good generalization at low temperatures is already announced in the high-temperature regime. Experimental bounds for single posterior draws, along with additional experiments including applications to model selection, are presented in Section D.2.

We provide a comparison of our results with the risk certificates of (Pérez-Ortiz et al., 2021). In Table 1, we compare our results on the full MNIST 10-class classification task against their certificates, using the same FCN and CNN architectures with a data-independent prior. For direct comparison, the baseline certificates for their method reported in Table 1 are taken from (Pérez-Ortiz et al., 2021, Table 1). Even though their method requires a specialized self-certified training algorithm, our approach yields tighter certificates for models achieving similar test error, without needing to change the standard training objective. Finally, we emphasize that these experiments evaluate our bounds on a multi-class classification task. This demonstrates the flexibility of our approach and its capability to scale beyond simple binary tasks into more practical settings.

*Table 1.* Train and test 01 errors and empirical true risk bounds, on MNIST for FCN and CNN architectures. The table contrasts our risk certificates for standard models with the self-certified networks of (Pérez-Ortiz et al., 2021) using a data-independent prior.

| Architecture | Method | Train 01 Error | Test 01 Error | True 01 Risk Bound |
|---|---|---|---|---|
| FCN | $f_{\text{quad}}$ | 0.0951 | 0.0921 | 0.3155 |
| | $f_{\text{lamda}}$ | 0.0742 | 0.0732 | 0.3275 |
| | $f_{\text{classic}}$ | 0.1531 | 0.1411 | 0.3304 |
| | $f_{\text{bbb}}$ | 0.0235 | **0.0293** | 0.5516 |
| | Ours | **0.0118** | 0.0338 | **0.2470** |
| CNN | $f_{\text{quad}}$ | 0.0535 | 0.0513 | 0.2165 |
| | $f_{\text{lamda}}$ | 0.0430 | 0.0397 | 0.2202 |
| | $f_{\text{classic}}$ | 0.0932 | 0.0869 | 0.2277 |
| | $f_{\text{bbb}}$ | 0.0120 | **0.0154** | 0.3645 |
| | Ours | **0.0032** | 0.0182 | **0.2025** |

As observed, our empirical results suggest that the main principle is not limited to Gibbs distributions, but also extends to nearby distributions approximated by LMC algorithms such as SGLD. Consequently, since Stochastic Gradient Descent (SGD) can be viewed as the zero-temperature limit of SGLD, our generalization guarantees at very low temperatures may potentially transfer to SGD. While our preliminary results in Table 3 in Section D.2.8 support this hypothesis, further empirical validation is required.

## 6. Conclusion

Using the integral representation of the log-partition function, the Gibbs posterior admits the computation of upper bounds on the true error based on the training data, and for any temperature. These bounds are stable under perturbation in relative entropy and can be approximated by Langevin Monte Carlo (LMC) algorithms. However, for realistic experiments, the approximations obtained by these algorithms are coarse and require calibration, which leads to rather tight bounds in the interpolation regime of overparametrized neural networks.

The fact that the calibrated bounds are very tight is, at this point, a purely experimental finding, requiring more theoretical investigation in future work.

## Impact Statement

This paper presents work whose goal is to advance the field of Machine Learning. There are many potential societal consequences of our work, none of which we feel must be specifically highlighted here.

ACKNOWLEDGMENTS

We acknowledge financial support from NextGenerationEU and MUR PNRR project PE0000013 CUP J53C22003010006 "Future Artificial Intelligence Research (FAIR)", and EU Project ELIAS (GA no 101120237). We would like to thank the CSML-IIT lab members for their valuable discussions and support during this research. In particular, we are especially grateful to Matia Bojovic and Alek Fröhlich for their time and help.

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

# Appendix

In this appendix, we provide a glossary of notation, give additional theoretical results and missing proofs, and provide more information on the numerical experiments, as well as additional experimental results.

## A. Table of notation

| Notation | Brief description | Section |
|---|---|---|
| $\mathcal{X}$ | space of data | 2, 3, 4 |
| $\Sigma$ | sigma algebra (events) on $\mathcal{X}$ | 2 |
| $\mu$ | probability of data | 2, 4.2 |
| $n$ | sample size | 1, 2, 3, 4 |
| $\delta$ | confidence parameter in high probability bounds | 1, 2, 4.2 |
| $\mathbf{x}$ | generic member $(x_1, ..., x_n) \in \mathcal{X}^n$, training sample | 1, 2, 3, 4 |
| $\mathcal{H}$ | hypothesis space | 2, 3, 4.2 |
| $\Omega$ | sigma algebra (events) on $\mathcal{H}$ | 2, 3 |
| $\ell$ | $\ell : \mathcal{H} \times \mathcal{X} \to [0, \infty)$ loss function | 1, 2, 4.2, 5 |
| $\mathcal{P}(\mathcal{H})$ | probability measures on $\mathcal{H}$ | 2, 3 |
| $\pi$ | nonnegative a-priori measure on $\mathcal{H}$ | 2, 3, 4 |
| $\|f\|_\infty$ | The sup-norm is defined: $\|f\|_\infty = \sup\{|f(s)| : s \in S\}$ | 2, 4 |
| $\sigma$ | width of Gaussian prior | 4.1, 5 |
| $L(h)$ | $L(h) = \mathbb{E}_{x \sim \mu}[\ell(h, x)]$, expected loss of $h \in \mathcal{H}$ | 2 |
| $\hat{L}(h, \mathbf{x})$ | $\hat{L}(h, \mathbf{x}) = (1/n) \sum_{i=1}^n \ell(h, x_i)$, empirical loss of $h \in \mathcal{H}$ | 2, 3, 4, 5 |
| $\beta$ | inverse temperature | 1, 2, 3, 4, 5 |
| $Z_\beta(\mathbf{x})$ | partition function | 3, 4.1 |
| $G_{\beta, \pi}(\mathbf{x})$ | Gibbs posterior with energy $\hat{L}$ and prior $\pi$ | 1, 3, 4, 5 |
| $\mathbb{E}_{g \sim G_\beta(\mathbf{x})}$ | posterior expectation | 3 |
| $\beta_0^K$ | increasing sequence $(0 = \beta_0 < \beta_1 < \cdots < \beta_K = \beta)$ of positive reals | 4.2, 5 |
| $\Gamma(\nu_0^{K-1}, \mathbf{x}, \beta_0^K)$ | bounding functional | 4.2, 5 |
| $F(h, \mathbf{x})$ | placeholder for generalization gap | 2, 4.2 |
| $\kappa(p, q)$ | $kl(p, q) = p \ln \frac{p}{q} + (1-p) \ln \frac{1-p}{1-q}$, rel. entropy of Bernoulli variables | 5 |
| $\kappa^{-1}(p, t)$ | $\inf\{q : q \geq p, \kappa(p, q) \geq t\}$ | 5.4 |
| $\mathrm{KL}(\rho, \nu)$ | $\int \left(\ln \frac{d\rho}{d\nu}\right) d\rho$, KL-divergence of $\rho, \nu \in \mathcal{P}(\mathcal{H})$ | 2, 3, 4, 5 |
| $R_\infty(\rho, \nu)$ | $\sup_{h \in \mathcal{H}} \ln \frac{d\rho}{d\nu}(h)$, Rényi-infinity divergence of $\rho, \nu \in \mathcal{P}(\mathcal{H})$ | 2, 4.2 |
| $d_{TV}(\rho, \nu)$ | total variation distance | C.4 |
| $W_p(\rho, \nu)$ | $p$-Wasserstein distance | C.4 |
| $\eta$ | step size or learning rate | C.1 |
| $\nu_{\beta, \eta}$ | invariant measure of LMC approximation of $G_\beta$ with step size $\eta$ | C.1 |
| $\nu_{\beta, \eta, t}$ | LMC approximation of $G_\beta$ with step size $\eta$ at iteration $t$ | C.1 |
| $\nu_{\beta, \epsilon}$ | invariant measure of CLD process with step size $\epsilon$ | C.1 |
| $\nu_{\beta, \epsilon, t}$ | CLD with step size $\epsilon$ at iteration $t$ | C.1 |
| $r$ | calibration factor | 5.4 |
| $\bar{\mathbf{x}}$ | randomly labeled data | 5.4 |
| $\mathbb{M}_{\mathrm{stop}}, \mathbb{M}_{\mathrm{erg}}$ | filters for stopping and ergodic mean | 5.3 |
| $L_{01}(h)$ | $L_{01}(h) = \mathbb{E}_{x \sim \mu}[\ell_{01}(h, x)]$, expected 01-loss of $h \in \mathcal{H}$ | 5.4 |
| $\hat{L}_{01}(h, \mathbf{x})$ | $\hat{L}(h, \mathbf{x}) = (1/n) \sum_{i=1}^n \ell_{01}(h, x_i)$, empirical 01-loss of $h \in \mathcal{H}$ | 5.4 |

## B. PAC-Bayes

We review PAC-Bayesian theory. There is no claim to novelty.

### B.1. Proof of the PAC-Bayesian Theorem

**Definition B.1.** Given a stochastic algorithm $\nu$ we define a probability measure $\rho_\nu$ on $\mathcal{H} \times \mathcal{X}^n$ by

$$\rho_\nu(A) = \mathbb{E}_{\mathbf{x} \sim \mu^n} \mathbb{E}_{h \sim \nu(\mathbf{x})} \left[1_A(h, \mathbf{x})\right] \quad \text{for} \ A \in \Omega \otimes \Sigma^{\otimes n}. \tag{9}$$

Then, $\mathbb{E}_{(h,\mathbf{x}) \sim \rho_\nu} [\phi(h, \mathbf{x})] = \mathbb{E}_{\mathbf{x}} \mathbb{E}_{h \sim \nu(\mathbf{x})} [\phi(h, \mathbf{x})]$ for measurable $\phi : \mathcal{H} \times \mathcal{X}^n \to \mathbb{R}$. To draw the pair $(h, \mathbf{x})$ from $\rho_\nu$ we first draw the training sample $\mathbf{x}$, and then sample $h$ from $\nu(\mathbf{x})$.

Restatement of Theorem 2.1

**Theorem B.2.** *Let $F : \mathcal{H} \times \mathcal{X}^n \to \mathbb{R}$ be some measurable function, and let $\nu$ be a stochastic algorithm such that $\nu(\mathbf{x})$ is absolutely continuous w.r.t. $\pi$ for all $\mathbf{x} \in \mathcal{X}^n$. Then*

*(i) for $\delta > 0$ with probability at least $1 - \delta$ in $\mathbf{x} \sim \mu^n$ and $h \sim \nu(\mathbf{x})$*

$$F(h, \mathbf{x}) \leq \ln \frac{d\nu(\mathbf{x})}{d\pi}(h) + \ln \mathbb{E}_{\mathbf{x}} \mathbb{E}_{g \sim \pi} \left[e^{F(g, \mathbf{x})}\right] + \ln \left(\frac{1}{\delta}\right)$$

*(ii) for $\delta > 0$ with probability at least $1 - \delta$ in $\mathbf{x} \sim \mu^n$*

$$\mathbb{E}_{h \sim \nu(\mathbf{x})} [F(h, \mathbf{x})] \leq KL(\nu(\mathbf{x}), \pi) + \ln \frac{\mathbb{E}_{\mathbf{x}} \mathbb{E}_{g \sim \pi} \left[e^{F(g, \mathbf{x})}\right]}{\delta}$$

*Proof.* By Markov's inequality, for any real random variable $Y$

$$\Pr\left\{Y > \ln \mathbb{E}\left[e^Y\right] + \ln(1/\delta)\right\} = \Pr\left\{e^Y > \mathbb{E}\left[e^Y\right]/\delta\right\} \leq \delta.$$

To prove (i), we apply this to the random variable $Y = F(h, \mathbf{x}) - \ln(d(\nu(\mathbf{x})/d\pi))(h)$ on the probability space $(\mathcal{H} \times \mathcal{X}^n, \Omega \otimes \Sigma^{\otimes n}, \rho_\nu)$. This gives, with probability at least $1 - \delta$ as $\mathbf{x} \sim \mu^n$ and $h \sim \nu(\mathbf{x})$),

$$F(h, \mathbf{x}) - \ln \frac{d\nu(\mathbf{x})}{d\pi}(h)$$
$$\leq \ln \mathbb{E}_{\mathbf{x}} \mathbb{E}_{g \sim \nu(\mathbf{x})} \left[e^{F(g, \mathbf{x}) - \ln(d(\nu(\mathbf{x})/d\pi))(g)}\right] + \ln(1/\delta)$$
$$= \ln \mathbb{E}_{\mathbf{x}} \mathbb{E}_{g \sim \pi} \left[e^{F(g, \mathbf{x}) - \ln(d(\nu(\mathbf{x})/d\pi))(g) + \ln(d(\nu(\mathbf{x})/d\pi))(g)}\right] + \ln(1/\delta)$$
$$= \ln \mathbb{E}_{\mathbf{x}} \mathbb{E}_{g \sim \pi} \left[e^{F(g, \mathbf{x})}\right] + \ln(1/\delta).$$

For (ii) apply Markov's inequality on the probability space $(\mathcal{X}^n, \Sigma^{\otimes n}, \mu^n)$ to the random variable $Y = \mathbb{E}_{g \sim \nu(\mathbf{x})}[F(g, \mathbf{x})] - KL(\nu(\mathbf{x}), \pi) = \mathbb{E}_{g \sim \nu(\mathbf{x})}[F(g, \mathbf{x}) - \ln(d(\nu(\mathbf{x})/d\pi))(g)]$ instead. By Jensen's inequality

$$e^{\mathbb{E}_{g \sim \nu(\mathbf{x})}[F(g, \mathbf{x}) - \ln(d(\nu(\mathbf{x})/d\pi))(g)]} \leq \mathbb{E}_{g \sim \nu(\mathbf{x})} \left[e^{F(g, \mathbf{x}) - \ln(d(\nu(\mathbf{x})/d\pi))(g)}\right].$$

Then proceed as before. $\square$

### B.2. Concrete PAC-Bayesian Bounds and the 0-1 Loss

Assume that $\ell$ has values in $[0, 1]$. To derive (2) from Theorem 2.1 let $F(h, \mathbf{x}) = n \kappa \left(\hat{L}(h, \mathbf{x}), L(h)\right)$, where $\kappa$ is the relative entropy of two Bernoulli variables with expectations $p$ and $q$

$$\kappa(p, q) = p \ln \frac{p}{q} + (1 - p) \ln \frac{1 - p}{1 - q}. \tag{10}$$

for which (Tolstikhin & Seldin, 2013) give the inversion rule $\kappa(p,q) \leq B \implies q - p \leq \sqrt{2pB} + 2B$. Tonelli's theorem and Theorem 1 of (Maurer, 2004) then give $\mathbb{E}_{\mathbf{x}}\mathbb{E}_{g\sim\pi}\left[e^{F(g,\mathbf{x})}\right] = \mathbb{E}_{g\sim\pi}\mathbb{E}_{\mathbf{x}}\left[e^{F(g,\mathbf{x})}\right] \leq 2\sqrt{n}$ for $n \geq 8$. Substitution in (i) of Theorem 2.1 and division by $n$ then gives with high probability as $\mathbf{x} \sim \mu^n$ and $h \sim \nu(\mathbf{x})$

$$\kappa\left(\hat{L}(h,\mathbf{x}), L(h)\right) \leq \frac{1}{n}\left(\ln\frac{d\nu(\mathbf{x})}{d\pi} + \ln\left(\frac{2\sqrt{n}}{\delta}\right)\right), \tag{11}$$

and the inversion rule of (Tolstikhin & Seldin, 2013) gives (2). The derivation of the bound for $\mathbb{E}_{h\sim\nu(\mathbf{x})}[L(h)]$ from Theorem 2.1 (ii) is analogous, we just have to use $\kappa\left(\mathbb{E}_{h\sim\nu(\mathbf{x})}\left[\hat{L}(h,\mathbf{x})\right], \mathbb{E}_{h\sim\nu(\mathbf{x})}[L(h)]\right) \leq \mathbb{E}_{h\sim\nu(\mathbf{x})}\kappa\left(\hat{L}(h,\mathbf{x}), L(h)\right)$ by the joint convexity of $\kappa$ (see (Cover, 1999)). This gives

$$\kappa\left(\mathbb{E}_{h\sim\nu(\mathbf{x})}\left[\hat{L}(h,\mathbf{x})\right], \mathbb{E}_{h\sim\nu(\mathbf{x})}[L(h)]\right) \leq \frac{1}{n}\left(KL(\nu(\mathbf{x}),\pi) + \ln\left(\frac{2\sqrt{n}}{\delta}\right)\right) \tag{12}$$

and the corresponding inequality obtained from the inversion rule.

The inversion rule produces directly interpretable bounds, but stronger is the direct inversion using the function $\kappa^1 : [0,1] \times [0,\infty)$ by

$$\kappa^{-1}(p,t) = \inf\{q : q \geq p, \kappa(p,q) \geq t\}.$$

In the definition of $F$, using the function $\kappa$, we can use other loss functions, possibly different from the loss function $\ell$ that defines the Gibbs posterior. If these loss functions satisfy the conditions of Theorem 1 in (Maurer, 2004), we obtain analogous bounds. In particular, for binary classification, we can use the 01 loss. Momentarily changing notation by replacing $x \in \mathcal{X}$ by $(x,y)$, where $y$ is the label corresponding to $x$, the 01 loss is defined as

$$\hat{L}_{\mathbf{01}}(h,\mathbf{x}) = \frac{1}{n}\sum_{i=1}^{n} 1_{(-\infty,0)}(y_i\ell(h,x_i))$$

and $L_{\mathbf{01}}(h) = \mathbb{E}_{x\sim\mu^n}\left[\hat{L}_{\mathbf{01}}(h,\mathbf{x})\right]$. We then obtain the bounds

$$\begin{aligned}
L_{\mathbf{01}}(h) &\leq \kappa^{-1}\left(\hat{L}_{\mathbf{01}}(h,\mathbf{x}), \frac{1}{n}\left(\ln\frac{d\nu(\mathbf{x})}{d\pi} + \ln\left(\frac{2\sqrt{n}}{\delta}\right)\right)\right) \\
\mathbb{E}_{h\sim\nu(\mathbf{x})}[L_{\mathbf{01}}(h)] &\leq \kappa^{-1}\left(\mathbb{E}_{h\sim\nu(\mathbf{x})}\left[\hat{L}_{\mathbf{01}}(h,\mathbf{x})\right], \frac{1}{n}\left(KL(\nu(\mathbf{x}),\pi) + \ln\left(\frac{2\sqrt{n}}{\delta}\right)\right)\right).
\end{aligned} \tag{13}$$

This is the bound used in our experiments.

## C. Supplementary material for Section 4

### C.1. Examples of LMC: CLD and ULA

A number of recent works give convergence guarantees for LMC algorithms and processes (Raginsky et al., 2017; Dalalyan & Karagulyan, 2019; Brosse et al., 2018; Vempala & Wibisono, 2019; Dwivedi et al., 2019; Nemeth & Fearnhead, 2021; Balasubramanian et al., 2022; Chen et al., 2022). Here we focus on the results of (Vempala & Wibisono, 2019), which do not require convexity of $V$ and instead assume that the measure $\nu$ satisfies a log-Sobolev inequality (LSI) in the sense that for all smooth $f : \mathbb{R}^d \to \mathbb{R}$

$$\mathbb{E}_{h\sim\nu}\left[f^2(h)\ln f^2(h)\right] - \mathbb{E}_{h\sim\nu}\left[f^2(h)\right]\ln\mathbb{E}_{h\sim\nu}\left[f^2(h)\right] \leq \frac{2}{\alpha}\mathbb{E}_{h\sim\nu}\left[\|\nabla f(h)\|^2\right] \tag{14}$$

for some $\alpha > 0$. An LSI is satisfied when $V$ is strongly convex, but, importantly, also for measures which are bounded perturbations of measures satisfying an LSI (Holley & Stroock, 1987). In our applications, this will be ensured for bounded losses, because of the Gaussian prior, but $\alpha$ will deteriorate as $\beta$ increases. Vempala & Wibisono (2019) give further examples and a list of references for measures that are not log-concave and satisfy an LSI. (Raginsky et al., 2017) show that under dissipativity conditions of the loss, the Gibbs posterior $G_\beta(\mathbf{x})$ satisfies an LSI with constant independent of $\mathbf{x}$. There are a number of more recent works on this topic, in particular proximal methods (see, e.g., (Chen et al., 2022)), but the work

of (Vempala & Wibisono, 2019) is convenient for our purposes, because they guarantee convergence in relative entropy, and our emphasis is not on sampling.

Continuous Langevin Dynamics (CLD) is specified by the stochastic differential equation

$$dh_t = -\nabla V(h_t)\, dt + \sqrt{2}dB_t,$$

where $B_t$ is centered standard Brownian motion in $\mathbb{R}^d$. The distribution of CLD converges exponentially to the Gibbs posterior under mild conditions (Chiang et al., 1987). In Section C.2, we give a convergence result for CLD adapted to temperature dependence and prior, on the condition of a log-Sobolev inequality, with convergence in relative entropy.

The Euler-discretization of CLD is the iterative algorithm

$$h_{t+1} = h_t - \epsilon \nabla V(h_t) + \sqrt{2\epsilon}\xi_t, \tag{15}$$

where $\epsilon > 0$ is a step size, the $\xi_t \sim \mathcal{N}(0, I)$ are independent Gaussian vectors and $h_0$ is drawn from some initial distribution $\nu_0$. Some authors call this algorithm simply LMC, for Langevin Monte Carlo. We call it ULA, alongside (Durmus & Moulines, 2017; Dwivedi et al., 2019; Vempala & Wibisono, 2019), for Unadjusted Langevin Algorithm. A popular variant of ULA is Stochastic Gradient Langevin Dynamics (SGLD) (Welling & Teh, 2011; Raginsky et al., 2017), where the gradient is replaced by an unbiased estimate, typically realized with random minibatches. Here, we restrict ourselves to ULA with a constant step size, because it has the fewest parameters to adjust, but in experiments, we also use the computationally more efficient SGLD.

The distribution $\nu_{\epsilon,t}$ of ULA converges as $t \to \infty$ to a biased limiting distribution $\nu_\epsilon$, which is generally different from $\nu$, but expected to be closer to $\nu$ as $\epsilon$ becomes smaller. (Vempala & Wibisono, 2019) use the LSI assumption to control the difference between CLD and ULA along their path and prove the following result.

**Theorem C.1.** *Assume that $\nu$ satisfies the log-Sobolev inequality (14) with $\alpha > 0$, that the Hessian of $V$ satisfies $-LI \preceq \nabla^2 V(h) \preceq LI$ for all $h$ and some $L < \infty$, and that $0 < \epsilon \le \alpha/\left(4L^2\right)$. Then, for $t \ge 0$*

$$KL(\nu_{\epsilon,t}, \nu) \le e^{-\alpha\epsilon t}KL(\nu_0, \nu) + \frac{8\epsilon dL^2}{\alpha}.$$

The first exponential term is due to the mismatch of the initial distribution and $\nu$. (Vempala & Wibisono, 2019) show that $\nu_0$ may be chosen to make KL$(\nu_0, \nu)$ of order $d$. The second term bounds the divergence between the limiting distribution $\nu_\epsilon$ and $\nu$. Similar results exist under different conditions on the potential $V$; (Cheng et al., 2018) for example, require $V$ to be strongly convex outside of a ball instead of the log-Sobolev inequality and give bounds in terms of the $W_1$-Wasserstein metric. (Raginsky et al., 2017) give bounds for $W_2$ under dissipativity assumptions. We are not aware of similar bounds for the Rényi-infinity divergence.

The next corollary adapts Theorem C.1 to the situation studied in this paper.

**Corollary C.2.** *For $\beta > 0$ consider the Gibbs posterior $G_\beta$ corresponding to $\hat{L}(h)$, with centered Gaussian prior of width $\sigma$. Assume that it satisfies the log-Sobolev inequality (14) with $\alpha > 0$, that the Hessian of $\hat{L}$ satisfies $-RI \preceq \nabla^2 \hat{L}(h) \preceq RI$ for all $h$ and some $R < \infty$, and that $0 < \eta \le \alpha/\left(4\left(\beta R + \frac{1}{\sigma^2}\right)^2\right)$. Consider the algorithm*

$$h_{t+1} = h_t - \eta\nabla_h \hat{L}(h_t) - \frac{\eta h_t}{\beta\sigma^2} + \sqrt{\frac{2\eta}{\beta}}\xi_t, \tag{16}$$

*where $h_0 \sim \nu_0$ and the $\xi_t \sim \mathcal{N}(0, I)$ are independent Gaussian random variables. Let $D(\beta) = KL(\nu_0, G_\beta)$ and let $\nu_{\beta,\eta,t}$ be the distribution of $h_t$ after $t$ steps. Then*

$$KL(\nu_{\beta,\eta,t}\|G_\beta) \le e^{-\frac{\alpha\eta t}{\beta}}D(\beta) + \frac{8\eta d}{\beta\alpha}\left(\beta R + \frac{1}{\sigma^2}\right)^2$$

*Proof.* This follows directly from Theorem C.1 and the substitutions $V(h) = \beta\hat{L}(h) + \|h\|^2/\left(2\sigma^2\right)$, $\epsilon = \eta/\beta$ and $L = \beta R + \frac{1}{\sigma^2}$. Then $\nu = G_\beta$ with Gaussian prior of width $\sigma$ and ULA becomes (16). $\square$

## C.2. Convergence of CLD under LSI

The following is a straightforward adaptation of Theorem 1 and its proof in (Vempala & Wibisono, 2019). There is no claim to originality.

**Lemma C.3.** *Let the process $h_{\beta,t}$ on $\mathbb{R}^d$ be defined by the stochastic differential equation*

$$dh_{\beta,t}(\mathbf{x}) = -\left(\nabla V(h_{\beta,t}) + \frac{h_{\beta,t}}{\beta\sigma^2}\right)dt + \sqrt{2/\beta}dB_t, \tag{17}$$

*and suppose that the measure with density $G_\beta(h) := \exp\left(-\left(\beta V + \frac{\|h\|^2}{2\sigma^2}\right)\right)$ satisfies an LSI with constant $\alpha$. Then, if $\nu_{\beta,t}$ is the distribution of $h_{\beta,t}$,*

$$KL(\nu_{\beta,t}, G_\beta) \le e^{-2\alpha t/\beta}KL(\nu_{\beta,0}, G_\beta).$$

*Proof.* Let $U_{\beta,t} = -\beta^{-1}\left(\ln\nu_{\beta,t} + \|h\|^2/2\sigma^2\right)$, so that $\nu_{\beta,t} = \exp\left(-\beta U_{\beta,t} - \frac{\|h\|^2}{2\sigma^2}\right)$ and

$$KL(\nu_{\beta,t}, G_\beta) = \beta\int_{\mathbb{R}^d}(V - U_{\beta,t})\nu_{\beta,t}d\lambda,$$

where $\lambda$ is Lebesgue measure on $\mathbb{R}^d$. The Fokker-Planck equation for (17) becomes

$$
\begin{aligned}
\frac{\partial\nu_{\beta,t}}{\partial t} &= \nabla\cdot\left(\nu_{\beta,t}\nabla\left(V + \frac{\|h\|^2}{2\beta\sigma^2}\right)\right) + \beta^{-1}\Delta\nu_{\beta,t} \\
&= \nabla\cdot(\nu_{\beta,t}\nabla(V - U_{\beta,t})).
\end{aligned}
$$

We have $0 = \frac{d}{dt}\int\nu_{\beta,t}d\lambda = -\int\left(\frac{\partial}{\partial t}\beta U_{\beta,t}\right)\nu_{\beta,t}dt$, so with integration by parts

$$
\begin{aligned}
\frac{d}{dt}KL(\nu_{\beta,t}, G_\beta) &= \int_{\mathbb{R}^d}\beta(V - U_{\beta,t})(\nabla\cdot(\nu_{\beta,s}\nabla(V - U_{\beta,t})))d\lambda \\
&= -\beta\int_{\mathbb{R}^d}\langle\nabla(V - U_{\beta,t}), \nabla(V - U_{\beta,t})\rangle\nu_{\beta,s}d\lambda \\
&= -\beta^{-1}\int_{\mathbb{R}^d}\|\nabla(\beta V - \beta U_{\beta,t})\|^2\nu_{\beta,t}d\lambda \\
&= -\beta^{-1}J(\nu_{\beta,t}, G_\beta) \\
&\le -2\alpha\beta^{-1}KL(\nu_{\beta,t}, G_\beta),
\end{aligned}
$$

where $J(\nu, \rho) = E_\nu\left\|\nabla\ln\frac{d\nu}{d\rho}\right\|^2$ is the relative Fisher information, and the last inequality follows from the LSI. Integrating this inequality concludes the proof. $\square$

## C.3. The "Second law of thermodynamics"

Restatement of Lemma 4.1

**Lemma C.4.** *Under the assumptions of Section 4.1, if $\nu$ is a stationary distribution of $\{h_t\}_{t\in I}$ and $s < t$ then $KL(\nu_t, \nu) \le KL(\nu_s, \nu)$ and $R_\infty(\nu_t, \nu) \le R_\infty(\nu_s, \nu)$, with equality in either case if and only if $\nu_s = \nu$.*

*Proof.* We identify measures with their densities with respect to the base distribution $\pi$, The transition kernel is denoted $K(h, g) = \mathbb{P}(h_{t-s} = h|h_0 = g) = \mathbb{P}(h_t = h|h_s = g)$.

$$
\begin{aligned}
\nu_t(h)\ln\frac{\nu_t(h)}{\nu(h)} &= \int K(h, g)\nu_s(g)d\pi(g)\ln\frac{\int K(h, g)\nu_s(g)d\pi(g)}{\int K(h, g)\nu(g)d\pi(g)} \\
&\le \int K(h, g)\nu_s(g)\ln\frac{\nu_s(g)}{\nu(g)}d\pi(g),
\end{aligned}\tag{18}
$$

with equality if and only if $\nu_s = \nu$. The first identity is owed to the invariance of $\nu$. The inequality is the log-sum inequality (Cover, 1999), followed by cancellation of $K(h, g)$. Then by Fubini's theorem

$$
\begin{aligned}
KL(\nu_t, \nu) &= \int \nu_t(h) \ln \frac{\nu_t(h)}{\nu(h)} d\pi(h) \\
&\leq \int \left( \int K(h, g) d\pi(h) \right) \nu_s(g) \ln \frac{\nu_s(g)}{\nu(g)} d\pi(g) \\
&= \int \nu_s(g) \ln \frac{\nu_s(g)}{\nu(g)} d\pi(g) = KL(\nu_s, \nu),
\end{aligned}
$$

which gives the first inequality. To prove the second inequality, divide (18) by $\nu_t(h)$ to get with Hölder's inequality

$$
\begin{aligned}
\ln \frac{\nu_t(h)}{\nu(h)} &\leq \frac{1}{\nu_t(h)} \int K(h, g) \nu_s(g) \ln \frac{\nu_s(g)}{\nu(g)} d\pi(g) \\
&\leq \frac{1}{\nu_t(h)} \int K(h, g) \nu_s(g) d\pi(g) \left( \sup_g \ln \frac{\nu_s(g)}{\nu(g)} \right) \\
&= R_\infty(\nu_s, \nu).
\end{aligned}
$$

Take the supremum in $h$ to get the second inequality. $\qquad\square$

## C.4. Proofs for Section 4.2

We assume that $\mathcal{H} = \mathbb{R}^d$ and prepare the proof of Lemma 4.4.

The total variation distance is defined as $d_{TV} : (\rho, \nu) \in \mathcal{P}(\mathcal{H}) \times \mathcal{P}(\mathcal{H}) \mapsto \sup_{A \in \Omega} |\rho(A) - \nu(A)|$. If $f$ is a bounded measurable function, then

$$
|\mathbb{E}_\rho[f] - \mathbb{E}_\nu[f]| \leq \|f\|_\infty d_{TV}(\rho, \nu).
$$

By Pinsker's inequality (see, e.g. (Boucheron et al., 2013))

$$
d_{TV}(\rho, \nu) \leq \sqrt{2KL(\rho, \nu)}.
$$

The $W_p$-Wasserstein distance is $W_p(\rho, \nu) = (\inf_W \mathbb{E}_{(x,y) \sim W}[\|x - y\|^p])^{1/p}$ with the infimum being over all probability measures on $\mathcal{P}(\mathcal{H} \times \mathcal{H})$ with $\rho$ and $\nu$ as marginals. We will use the following fact: Since $W_1 \leq W_2$ it follows from the Kantorovich-Rubinstein Theorem (Villani, 2009), that for any real Lipschitz function $f$ on $\mathcal{H}$ and probability measures $\nu_1, \nu_2 \in \mathcal{P}(\mathcal{H})$

$$
|\mathbb{E}_{h \sim \nu_1}[f(h)] - \mathbb{E}_{h \sim \nu_2}[f(h)]| \leq \|f\|_{\mathrm{Lip}} W_1(\nu_1, \nu_2) \leq \|f\|_{\mathrm{Lip}} W_2(\nu_1, \nu_2),
$$

where $\|.\|_{\mathrm{Lip}}$ is the Lipschitz-seminorm, $\|f\|_{Lip} = \inf \{ s : f(h) - f(g) \leq s \|h - g\| \text{ for all } h, g \in \mathbb{R}^d \}$. If $\nu$ satisfies an LSI with constant $\alpha$ as in (14), then (Otto & Villani, 2000)

$$
W_p(\rho, \nu) \leq \frac{2}{\alpha} KL(\rho, \nu).
$$

Restatement of Lemma 4.4.

**Lemma C.5.** *Denote*

$$
\Delta := \int_0^\beta \mathbb{E}_{h \sim G_\gamma(\mathbf{x})} \left[ \hat{L}(h, \mathbf{x}) \right] d\gamma - \Gamma(\nu_0^{K-1}, \mathbf{x}, \beta_0^K)
$$

*(i) If $\mathbb{E}_{h \sim G_{\beta_k}(\mathbf{x})} \left[ \hat{L}(h, \mathbf{x}) \right] \leq \mathbb{E}_{h \sim \nu_k(\mathbf{x})} \left[ \hat{L}(h, \mathbf{x}) \right]$ for all $k$ and $\mathbf{x}$, then $\Delta \leq 0$.*

*(ii) If $\ell(h, \mathbf{x})$ is bounded in $h$ for all $\mathbf{x}$, $\|\ell(h, \mathbf{x})\| \leq m$ then*

$$
\Delta \leq m \sum_{k=1}^K (\beta_k - \beta_{k-1}) \sqrt{KL(\nu_{k-1}(\mathbf{x}), G_{\beta_{k-1}}(\mathbf{x}))/2}.
$$

*(iii) If instead $\ell(h, \mathbf{x})$ is $m$-Lipschitz in $h$ for all $\mathbf{x}$, $\ell(h, \mathbf{x}) - \ell(g, \mathbf{x}) \le m \|h - g\|$ and $G_{\beta_k}(\mathbf{x})$ satisfies an LSI with constant $\alpha$ for all $k$ and $\mathbf{x}$, then*

$$\Delta \le \frac{2m}{\alpha} \sum_{k=1}^{K} (\beta_k - \beta_{k-1}) KL\left(\nu_{k-1}(\mathbf{x}), G_{\beta_{k-1}}(\mathbf{x})\right).$$

*Proof.* By the last assertion of Lemma 3.1 the function $\beta \mapsto \mathbb{E}_{g \sim G_\beta(\mathbf{x})}[\hat{L}(g, \mathbf{x})]$ is non-increasing, so

$$\begin{aligned}
\Delta &= \int_0^\beta \mathbb{E}_{h \sim G_\gamma(\mathbf{x})}\left[\hat{L}(h, \mathbf{x})\right] d\gamma - \Gamma(\nu_0^{K-1}, \mathbf{x}, \beta_0^K) \\
&\le \sum_{k=1}^{K} (\beta_k - \beta_{k-1}) \mathbb{E}_{h \sim G_{\beta_{k-1}}(\mathbf{x})}\left[\hat{L}(h, \mathbf{x})\right] - \Gamma(\nu_0^{K-1}, \mathbf{x}, \beta_0^K) \\
&= \sum_{k=1}^{K} (\beta_k - \beta_{k-1}) \left(\mathbb{E}_{h \sim G_{\beta_{k-1}}(\mathbf{x})}\left[\hat{L}(h, \mathbf{x})\right] - \mathbb{E}_{h \sim \nu_{k-1}(\mathbf{x})}\left[\hat{L}(h, \mathbf{x})\right]\right).
\end{aligned}$$

Now if $\mathbb{E}_{h \sim G_{\beta_{k-1}}(\mathbf{x})}\left[\hat{L}(h, \mathbf{x})\right] \le \mathbb{E}_{h \sim \nu_{k-1}(\mathbf{x})}\left[\hat{L}(h, \mathbf{x})\right]$ then (i) is immediate. If $\ell$ is bounded by $m$ then $\left\|\hat{L}(., \mathbf{x})\right\|_\infty \le m$ and

$$\left(\mathbb{E}_{h \sim G_{\beta_{k-1}}(\mathbf{x})} - \mathbb{E}_{h \sim \nu_{k-1}(\mathbf{x})}\right)\left[\hat{L}(h, \mathbf{x})\right] \le m \, d_{TV}\left(G_{\beta_{k-1}}(\mathbf{x}), \nu_{k-1}(\mathbf{x})\right) \le m\sqrt{KL\left(\nu_{k-1}, G_{\beta_{k-1}}\right)/2},$$

by Pinsker's inequality, which gives (ii). If $\ell$ is Lipschitz in the 1st argument with constant $m$, then so is $\hat{L}$, and

$$\left(\mathbb{E}_{h \sim G_{\beta_{k-1}}(\mathbf{x})} - \mathbb{E}_{h \sim \nu_{k-1}(\mathbf{x})}\right)\left[\hat{L}(h, \mathbf{x})\right] \le m \, W_2\left(G_{\beta_{k-1}}(\mathbf{x}), \nu_{k-1}(\mathbf{x})\right) \le 2mKL\left(\nu_{k-1}, G_{\beta_{k-1}}\right)/\alpha,$$

which gives (ii). $\qquad\square$

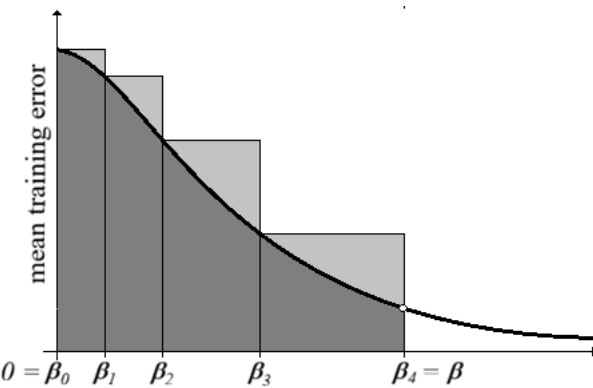

*Figure 5.* An illustration of the computation of the functional $\Gamma(\nu_0^{K-1}, \mathbf{x}, \beta_0^K)$.

Restatement of Theorem 4.5:

**Theorem C.6.** *Let $F : \mathcal{H} \times \mathcal{X}^n \to \mathbb{R}$ be some measurable function and $\beta_0^K$ and $\nu_0^{K-1}$ as in Definition 4.3. Let $\nu(\mathbf{x})$ be any data-dependent distribution on $\mathcal{H}$. Let $\Delta$ be bounded as in Lemma 4.4, depending on which of the conditions is fulfilled by $\ell$. Then*

*(i) with probability at least $1 - \delta$ as $\mathbf{x} \sim \mu^n$ and $h \sim \nu(\mathbf{x})$*

$$\begin{aligned}
F(h, \mathbf{x}) &\le -\beta \hat{L}(h, \mathbf{x}) + \Gamma(\nu_0^{K-1}, \mathbf{x}, \beta_0^K) + \ln \mathbb{E}_{\mathbf{x}} \mathbb{E}_{h \sim \pi}\left[e^{F(h, \mathbf{x})}\right] + \ln \frac{1}{\delta} \\
&\quad + R_\infty\left(\nu(\mathbf{x}), G_\beta(\mathbf{x})\right) + \Delta.
\end{aligned}$$

*If $F$ and $\ell$ are bounded, then $R_\infty\left(\nu\left(\mathbf{x}\right), G_\beta\left(\mathbf{x}\right)\right)$ can be replaced by*

$$\max\left\{0, \beta\left\|\ell\right\|_\infty + \left\|F\right\|_\infty + \ln\sqrt{2KL\left(\nu\left(\mathbf{x}\right), G_\beta\left(\mathbf{x}\right)\right)}\right\}.$$

*(ii) with probability at least $1 - \delta$ as $\mathbf{x} \sim \mu^n$*

$$\mathbb{E}_{\nu(\mathbf{x})}[F(h,\mathbf{x})] \leq -\beta\mathbb{E}_{\nu(\mathbf{x})}\big[\hat{L}\left(h,\mathbf{x}\right)\big] + \Gamma(\nu_0^{K-1}, \mathbf{x}, \beta_0^K) + \ln\mathbb{E}_\mathbf{x}\mathbb{E}_{h\sim\pi}\left[e^{F(h,\mathbf{x})}\right] + \ln\frac{1}{\delta}$$
$$+ KL\left(\nu\left(\mathbf{x}\right), G_\beta\left(\mathbf{x}\right)\right) + \Delta.$$

*Proof.* Proof of (ii). By equation (6) and Lemma 4.4 we have

$$
\begin{aligned}
KL\left(\nu\left(\mathbf{x}\right), \pi\right) &\leq KL\left(\nu\left(\mathbf{x}\right), G_\beta\left(\mathbf{x}\right)\right) - \beta\mathbb{E}_{h\sim\nu(\mathbf{x})}\left[\hat{L}\left(h,\mathbf{x}\right)\right] + \int_0^\beta \mathbb{E}_{h\sim G_\gamma(\mathbf{x})}d\gamma \\
&\leq KL\left(\nu\left(\mathbf{x}\right), G_\beta\left(\mathbf{x}\right)\right) - \beta\mathbb{E}_{h\sim\nu(\mathbf{x})}\left[\hat{L}\left(h,\mathbf{x}\right)\right] + \Gamma(\nu_0^{K-1}, \mathbf{x}, \beta_0^K) + \Delta.
\end{aligned}
$$

Substitution in (ii) of the PAC-Bayesian theorem proves (ii).

Proof of (i) without amendment. Similarly, we have from Lemma 4.4

$$
\begin{aligned}
\ln\frac{d\nu\left(\mathbf{x}\right)}{d\pi}\left(h\right) &= \ln\frac{d\nu\left(\mathbf{x}\right)}{dG_\beta\left(\mathbf{x}\right)}\left(h\right) + \ln\frac{dG_\beta\left(\mathbf{x}\right)}{d\pi}\left(h\right) \\
&\leq R_\infty\left(\nu\left(\mathbf{x}\right), G_\beta\left(\mathbf{x}\right)\right) - \beta\hat{L}\left(h,\mathbf{x}\right) + \int_0^\beta \mathbb{E}_{h\sim G_\gamma(\mathbf{x})}d\gamma \\
&= R_\infty\left(\nu\left(\mathbf{x}\right), G_\beta\left(\mathbf{x}\right)\right) - \beta\hat{L}\left(h,\mathbf{x}\right) + \Gamma(\nu_0^{K-1}, \mathbf{x}, \beta_0^K) + \Delta.
\end{aligned}
$$

Then substitute in the PAC-Bayesian theorem (i).

For the proof of the amendment to (i), we have to return to the proof of the PAC-Bayesian theorem. From Markov's inequality, we have (in analogy to the proof of Theorem 2.1) with probability at least $1 - \delta$ as $x \sim \mu^n$ and $h \sim \nu\left(\mathbf{x}\right)$, that

$$
\begin{aligned}
&F\left(h,\mathbf{x}\right) + \beta\hat{L}\left(h,\mathbf{x}\right) + \ln Z_\beta\left(\mathbf{x}\right) \\
\leq\ & \ln\mathbb{E}_\mathbf{x}\mathbb{E}_{h\sim\nu(\mathbf{x})}\left[e^{F(h,\mathbf{x})+\beta\hat{L}(h,\mathbf{x})+\ln Z_\beta(\mathbf{x})}\right] + \ln\left(1/\delta\right) \\
\leq\ & \ln\left(\mathbb{E}_\mathbf{x}\mathbb{E}_{h\sim G_\beta(\mathbf{x})}\left[e^{F(h,\mathbf{x})+\beta\hat{L}(h,\mathbf{x})+\ln Z_\beta(\mathbf{x})}\right] + e^{\beta\|\ell\|_\infty+\|F\|_\infty}d_{TV}\left(\nu\left(\mathbf{x}\right), G_\beta\left(\mathbf{x}\right)\right)\right) + \ln\left(1/\delta\right) \\
=\ & \ln\left(\max\left\{\mathbb{E}_\mathbf{x}\mathbb{E}_{h\sim\pi}\left[e^{F(h,\mathbf{x})}\right], 1\right\} + \max\left\{e^{\beta\|\ell\|_\infty+\|F\|_\infty}d_{TV}\left(\nu\left(\mathbf{x}\right), G_\beta\left(\mathbf{x}\right)\right), 1\right\}\right) + \ln\left(1/\delta\right) \\
\leq\ & \max\left\{\ln\mathbb{E}_\mathbf{x}\mathbb{E}_{h\sim\pi}\left[e^{F(h,\mathbf{x})}\right], 0\right\} + \max\left\{\beta\|\ell\|_\infty + \|F\|_\infty + \ln\left(2d_{TV}\left(\nu\left(\mathbf{x}\right), G_\beta\left(\mathbf{x}\right)\right)\right), 0\right\} \\
& + \ln\left(1/\delta\right).
\end{aligned}
$$

In the second inequality we used $\ln Z_\beta\left(\mathbf{x}\right) \leq 0$ and the property of the total variation metric. In the next inequality we used Lemma 3.1, and in the last line we used for $a, b \geq 1$ that $\ln\left(a + b\right) \leq \ln\max\left\{a, b\right\} + \ln 2 \leq \ln a + \ln b + \ln 2$. Subtract $\beta\hat{L}\left(h,\mathbf{x}\right) + \ln Z_\beta\left(\mathbf{x}\right)$, use Lemma 3.1, Lemma 4.4, and Pinsker's inequality to bound the total variation distance.

$\square$

## C.5. Miscellaneous Lemmata

**Lemma C.7.** *For $0 < \beta < \infty$*

$$\max\left\{KL\left(G_\beta, G_{2\beta}\right), KL\left(G_{2\beta}, G_{2\beta}\right)\right\} \leq \beta\left(\mathbb{E}_{h\sim G_\beta}\left[\hat{L}\left(h\right)\right] - \mathbb{E}_{h\sim G_{2\beta}}\left[\hat{L}\left(h\right)\right]\right)$$

*Proof.*

$$
\begin{aligned}
KL\left(G_\beta, G_{2\beta}\right) &= \mathbb{E}_{h\sim G_\beta}\left[-\beta\hat{L}\left(h\right) - \ln Z_\beta + 2\beta\hat{L}\left(h\right) + \ln Z_{2\beta}\right] \\
&= \mathbb{E}_{h\sim G_\beta}\left[\beta\hat{L}\left(h\right)\right] - \int_\beta^{2\beta}\mathbb{E}_{h\sim G_\gamma}\left[\hat{L}\left(h\right)\right]d\gamma \\
&\leq \beta\left(\mathbb{E}_{h\sim G_\beta}\left[\hat{L}\left(h\right)\right] - \mathbb{E}_{h\sim G_{2\beta}}\left[\hat{L}\left(h\right)\right]\right)
\end{aligned}
$$

Similarly,

$$
\begin{aligned}
KL\left(G_{2\beta}, G_\beta\right) &= \mathbb{E}_{h\sim G_{2\beta}}\left[-2\beta\hat{L}\left(h\right) - \ln Z_{2\beta} + \beta\hat{L}\left(h\right) + \ln Z_\beta\right] \\
&= -\mathbb{E}_{h\sim G_\beta}\left[\beta\hat{L}\left(h\right)\right] + \int_\beta^{2\beta}\mathbb{E}_{h\sim G_\gamma}\left[\hat{L}\left(h\right)\right]d\gamma \\
&\leq \beta\left(\mathbb{E}_{h\sim G_\beta}\left[\hat{L}\left(h\right)\right] - \mathbb{E}_{h\sim G_{2\beta}}\left[\hat{L}\left(h\right)\right]\right)
\end{aligned}
$$

$\square$

## C.6. The Calibration Factor

Let $\bar{A} = -\beta\mathbb{E}_{h\sim\nu_\beta}\left[\hat{L}\left(h, \bar{\mathbf{x}}\right)\right] + \Gamma\left(\nu_0^{K-1}, \bar{\mathbf{x}}, \beta_0^K\right)$ be the area estimate obtained for the random labels (corrected by $-\beta\mathbb{E}_{h\sim\nu_\beta}\left[\hat{L}\left(h, \bar{\mathbf{x}}\right)\right]$, which should play little role for large $\beta > n$). Then

$$
r\left(\mathbf{x}\right) = \min\left\{r : \forall k \in [K], \ \kappa^{-1}\left(\mathbb{E}_{h\sim\nu_{\beta_k}(\mathbf{x})}\left[\hat{L}_{01}\left(h, \bar{\mathbf{x}}\right)\right], \frac{1}{n}\left(r\bar{A} + \ln\frac{2\sqrt{n}}{\delta}\right)\right) \geq \frac{1}{2}\right\},
$$

where $\bar{\mathbf{x}}$ is the training set $\mathbf{x}$ with random labels and $\hat{L}_{01}$ the empirical 01-error. The calibration value $r$ is thus the smallest factor of $\bar{A}$, for which we obtain a correct upper bound on the 01-error with random labels for all the $\beta_k$.

## C.7. Calibration Justification

Let's assume the Bayesian Linear regression setting, where we are considering a quadratic loss function with an isotropic Gaussian prior. It will be easy to verify that the Gibbs posterior is Gaussian. Wibisono (2018) in Example 2.1 shows that the stationary distribution of ULA is also Gaussian with the same mean but inflated variance. In this setting, both the KL divergence (between ULA's invariant distribution and the prior) and the integral term derived from Lemma 3.1 can be computed analytically. Crucially, the label dependency appears only in a constant term, which becomes negligible in the low-temperature regime. Therefore, our assumed ratio is theoretically correct. The same argument shows that ULA, due to its inflation in estimating covariance, overestimates the mean training error relative to the correct distribution. Furthermore, assuming the ULA approximation bound by (Vempala & Wibisono, 2019) is sharp, similar reasoning demonstrates this ratio is fixed for Generalized Linear Models (GLMs) and the Neural Tangent Kernel (NTK) regime. The more detailed version of this argument will be provided in the extended version of the paper.

## D. Experimental Details and Additional Results

### D.1. Experimental Details

All the codes to reproduce the results are provided through this https://github.com/erfunmirzaei/Gibbs-Generalization. For all the experiments, we use an isotropic Gaussian prior with $\mu = 0$, for bounded loss with $\sigma = 5$ and for unbounded loss with $\sigma = 0.1$. This induces an L2-regularization term in the energy function that is stated in the proof of Corollary C.2. The confidence parameter $\delta$ appearing in our bounds is set to 0.01 for all experiments

We use either standard SGLD or ULA with a constant step size and without additional correction terms. When ULA has been used, we use a step size of 0.01 for both datasets. However, with SGLD, we set the step size to 0.01 for MNIST and 0.005 for CIFAR-10. For both datasets, MNIST and CIFAR-10, we use neural networks with ReLU activation functions.

### D.1.1. NETWORKS ARCHITECTURE

The fully connected networks consist of one, two, or three hidden layers, each containing a constant number of units. Besides that, we are using the LeNet-5 architecture for MNIST and the VGG16 architecture for CIFAR-10 to achieve low test error. For loss function $\ell$, we are mostly using bounded loss functions such as bounded binary cross-entropy (BBCE) as described in Appendix D of (Dziugaite & Roy, 2018) or the Savage loss (Masnadi-Shirazi & Vasconcelos, 2008). As an unbounded loss function, we tried binary cross-entropy (BCE) (Section D.2.6), but with a smaller value of $\sigma$, so as to avoid excessive training errors for small values of $\beta$.

The LeNet-5 network follows a systematic pattern of alternating convolutional and pooling layers, followed by fully connected layers (LeCun et al., 2002). It begins with an input layer that accepts $32 \times 32$ grayscale images. Thus, we pad our images to fit. The first convolutional layer (C1) applies 6 filters of size $5 \times 5$ to extract low-level features, followed by a $2 \times 2$ average pooling layer (S2) for spatial downsampling. The second convolutional layer (C3) uses 16 filters of size $5 \times 5$ to capture more complex feature combinations, followed again by a $2 \times 2$ average pooling layer (S4). A third convolutional layer (C5) with 120 filters of size $5 \times 5$ acts as a feature extractor, producing 120 feature maps, each of size $1 \times 1$. The architecture concludes with two fully connected layers: F6 with 84 neurons and a final output layer with 10 neurons for the original digit classification task. However, for our binary classification task, we modify F6 to have 420 neurons and use a single-neuron output layer. Throughout the network, ReLU activation functions replace the original tanh activations, which improves gradient flow and training performance in modern implementations.

VGG-16 is a widely used deep convolutional neural network architecture known for its simplicity and strong performance in image classification tasks (Simonyan & Zisserman, 2014). The architecture follows a consistent design using only $3 \times 3$ convolutional filters and $2 \times 2$ max pooling operations throughout the network. In our implementation, VGG-16 is adapted to handle CIFAR-10's smaller $32 \times 32$ RGB images. The network consists of 13 convolutional layers organized into five blocks: the first two blocks contain two convolutional layers each with 64 and 128 filters, respectively, while the last three blocks contain three convolutional layers each with 256, 512, and 512 filters, respectively. Each block is followed by a $2 \times 2$ max pooling layer for spatial downsampling. All convolutional layers employ $3 \times 3$ kernels with padding to preserve spatial dimensions, and ReLU activation functions introduce non-linearity. The convolutional feature extractor is followed by a classifier head consisting of three fully connected layers: two hidden layers with 1024 neurons each, using ReLU activation, and a final output layer with 1 neuron for binary classification. We also removed dropout to ensure that SGLD minimizes the defined energy function without any additional terms.

For MNIST, the input is a 784-dimensional vector, and the output is a scalar since we perform binary classification between digits 0–4 and 5–9. For CIFAR-10, the input dimension is 3072, and the output is again scalar, corresponding to binary classification between vehicles and animals. For evaluating our models, we are using all 10,000 test examples for both datasets.

### D.1.2. MINIBATCHES

When using SGLD, we adopt minibatches of size proportional to $\sqrt{n}$. Thus, for $n = 2000$ the minibatch size is 50, and for $n = 8000$ it is 100.

### D.1.3. MOVING AVERAGE FILTERS

As we explained in Section 5.3, we are using a running mean $\mathbb{M}(x_1, \cdots, x_t)$ of $\hat{L}(h_j, \mathbf{x})$ from $j = 1, \cdots, t$ both as a criterion to stop the experiment and an estimation for $\mathbb{E}_{h \sim G_{\beta_k}} \left[ \hat{L}(h, \mathbf{x}) \right]$. We define the running mean recursively in one of two ways:

$$\mathbb{M}_t = \tfrac{\alpha}{2} \hat{L}(h_t, \mathbf{x}) + \tfrac{\alpha}{2} \hat{L}(h_{t-1}, \mathbf{x}) + (1 - \alpha) \mathbb{M}_{t-1},$$
$$\mathbb{M}_t = \alpha \hat{L}(h_t, \mathbf{x}) + (1 - \alpha) \mathbb{M}_{t-1},$$

with $\mathbb{M}_0 = 1$ and small $\alpha$. We use the first (symmetric) form in the experiments with ULA, and the second (standard exponential moving average) form with SGLD for convenience. We set different values of $\alpha$ for the two roles: $\alpha = 0.0025$ for the stopping criterion ($\mathbb{M}_{\text{stop}}$) and $\alpha = 0.01$ for approximating the ergodic mean ($\mathbb{M}_{\text{erg}}$). The stopping rule is triggered when

$$\mathbb{M}_t - \mathbb{M}_{t-1} \geq \epsilon,$$

with $\epsilon = 10^{-7}$. To avoid premature termination, we impose a minimum of 4000 steps before applying this criterion. As $\alpha \to 0$ and $t \to \infty$, the quantity $\mathbb{M}_t$ converges to the ergodic mean.

## D.2. Experimental Results

### D.2.1. ILLUSTRATION OF BOUND COMPUTATION

In this section, we discuss Figure 4 in the main body in more detail. Figure 6 illustrates how our bounds are computed. The sequence of mean training losses in $\ell$ is used to compute for each $\beta$ the functional $\Gamma$ and the "KL-Bound", which corresponds to the right-hand side of the inequalities in Corollary 12. Our bound on the test loss is then computed by applying the function $\kappa^{-1}$ to the empirical 0-1 error and to this kl-bound. The graph of "KL(Train, Test)" corresponds to the left-hand side in Corollary 12. Remarkably, the close fit of the upper bound on the random labels is achieved by the adjustment of a single calibration parameter.

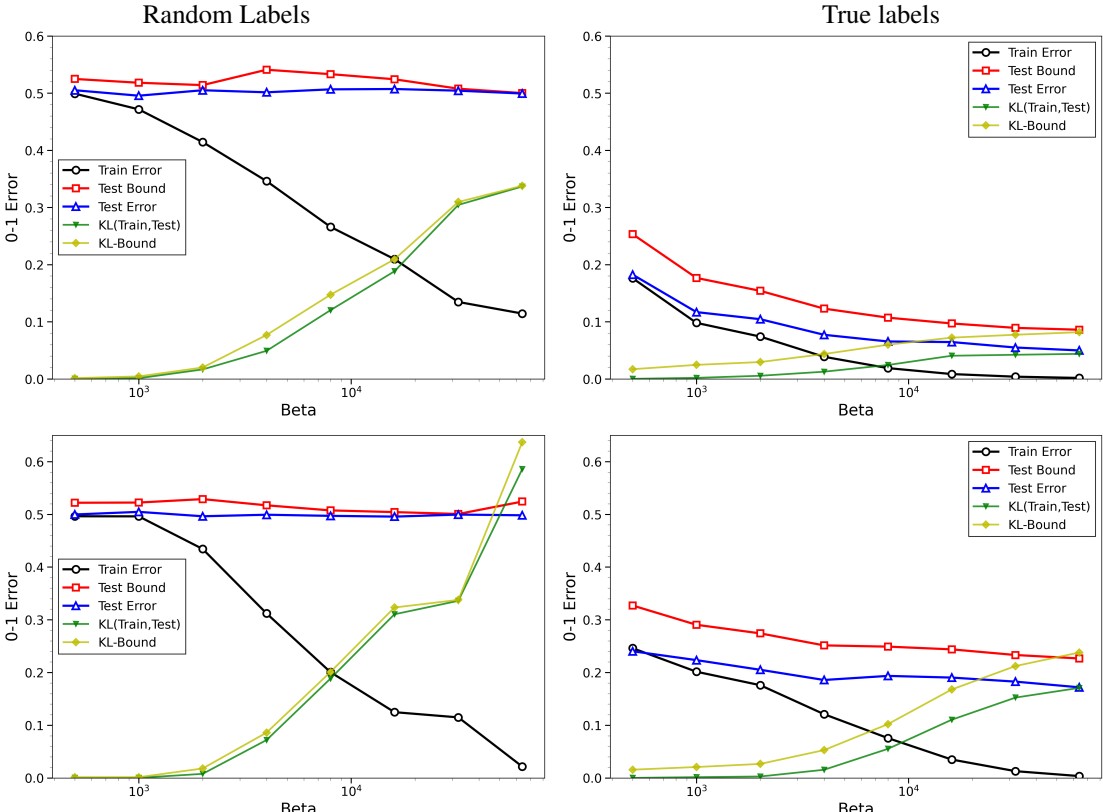

*Figure 6.* A more detailed version of Figure 4 to illustrate how the bounds are computed.

### D.2.2. SINGLE-DRAWS

For the setting described in Section 5.5, we also present the bounds for the single-draw case in Figure 7. It is noteworthy that, although the theoretical guarantees for this scenario are rather weak, the empirical bounds behave well. However, as visible in the plots, the results exhibit fluctuations and irregularities caused by stochastic effects, which make them less reliable.

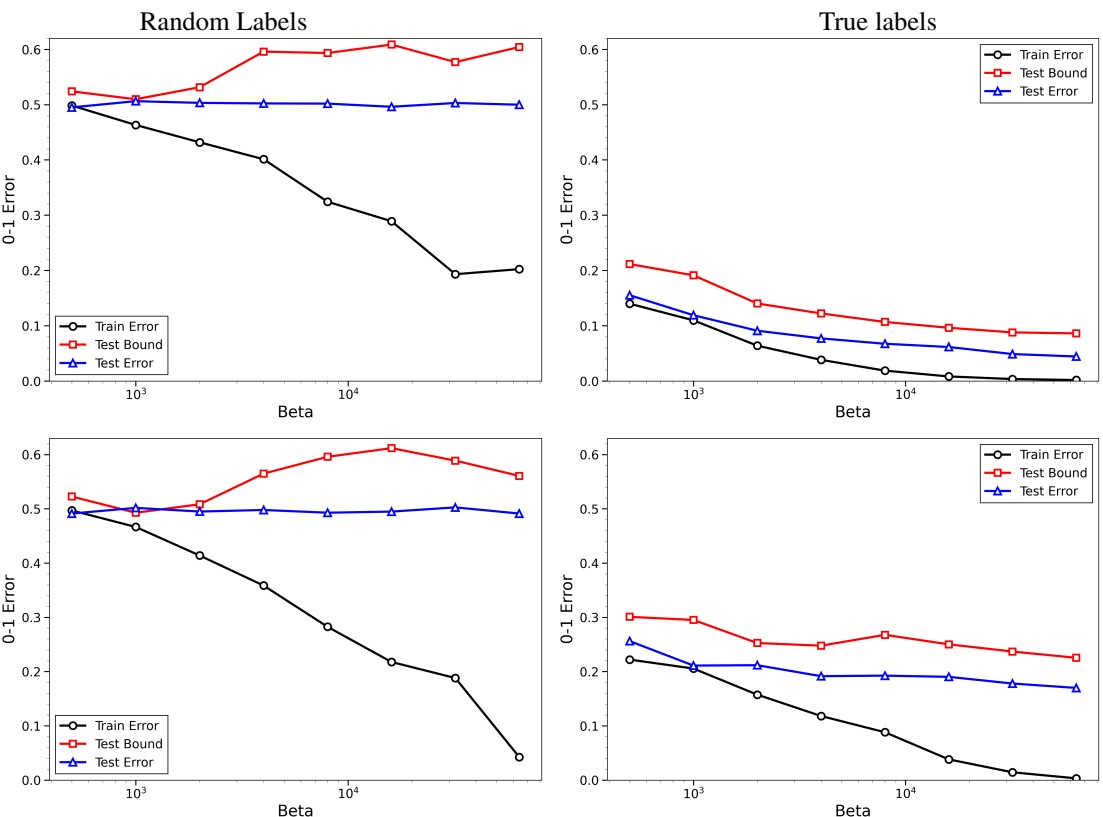

*Figure 7.* SGLD on MNIST and CIFAR-10 with 8000 training examples using the BBCE loss function. The first row corresponds to MNIST and the second row to CIFAR-10. Random labels are shown on the left, true labels on the right. Both random and true labels are trained with exactly the same algorithm and parameters on a fully connected ReLU network with two hidden layers of 1000 (respectively 1500) units. The calibration factor for MNIST is 0.77, for CIFAR-10, 0.89. Train error, test error, and our bound for a single draw of the 0-1 loss are plotted against $\beta$.

### D.2.3. DIFFERENT ARCHITECTURES

In this section, we evaluate the performance of different models and architectures on both MNIST and CIFAR-10, demonstrating that our bound can be used to guide model selection. In addition to the two-hidden-layer neural networks described in Section 4, we consider fully connected neural networks with three hidden layers, containing 500 and 1000 units for MNIST and CIFAR-10, respectively. Furthermore, we employ the LeNet-5 architecture for MNIST and the VGG-16 for CIFAR-10 to achieve high test accuracy. Detailed descriptions of these architectures are provided in Section D.1.1.

Figure 8 demonstrates the robustness of our bound across different models. We observe that the bounds can be very tight even when the test error is small. For convolutional neural networks, especially on the MNIST dataset, we observe strong performance with the true labels, but relatively poor performance with random labels, despite having more parameters than training examples. This can be explained by the fact that convolutional architectures are still far from being highly overparameterized. For the MNIST dataset, we use fully connected neural networks with two or three hidden layers, containing 1000 or 500 units per layer, respectively. This corresponds to a total of approximately 1,787,000 and 893,000 parameters, resulting in a parameter-to-training-example ratio of roughly 200 and 100, respectively. In contrast, LeNet-5 has around 100,000 parameters, yielding a ratio of approximately 12.5.

The empirical test bounds can serve as a selection criterion among different models. Table 2 show that test bounds at low temperature are useful for model selection, and that bounds at high temperature can also predict the behavior of the model at low temperature.

|  | 2HL (W=1000) | 3HL (W=500) | LeNet-5 |
| --- | --- | --- | --- |
| Test Bound at $\beta = 1k$ | 0.1766 | 0.2347 | 0.0887 |
| Test Error at $\beta = 64k$ | 0.0498 | 0.0549 | 0.0317 |
| Test Bound at $\beta = 64k$ | 0.0860 | 0.1314 | 0.0375 |

*(a)* MNIST, 8k training examples (true labels).

|  | 2HL (W=1500) | 3HL (W=1000) | VGG-16 |
| --- | --- | --- | --- |
| Test Bound at $\beta = 1k$ | 0.2905 | 0.3635 | 0.2330 |
| Test Error at $\beta = 64k$ | 0.1719 | 0.1782 | 0.0903 |
| Test Bound at $\beta = 64k$ | 0.2266 | 0.2807 | 0.2030 |

*(b)* CIFAR-10, 8k training examples (true labels).

*Table 2.* Test bounds and test errors for different neural network architectures on MNIST and CIFAR-10. The bounds at both low and high temperatures reliably reflect test error performance at low temperature.

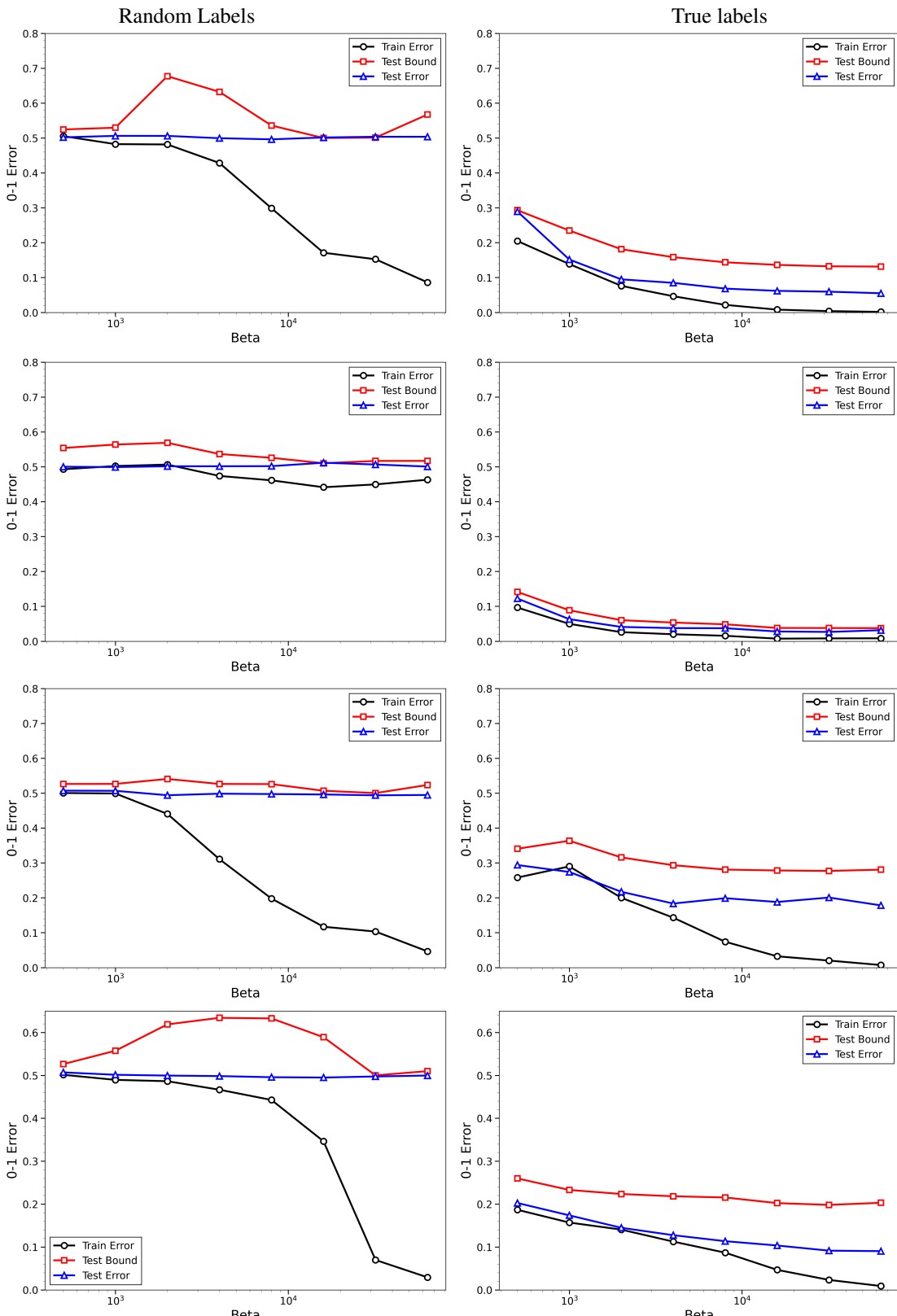

*Figure 8.* SGLD on MNIST and CIFAR-10 with 8000 training examples using the BBCE loss function. The first two rows correspond to MNIST, and the remaining rows to CIFAR-10. Random labels are shown on the left, and true labels on the right. Both random and true labels are trained using the same algorithm and hyperparameters on a fully connected ReLU network with three hidden layers of 500 (MNIST) or 1000 (CIFAR-10) units, followed by LeNet-5 (MNIST) or VGG-16 (CIFAR-10), shown in the subsequent row. The calibration factors for MNIST are 0.26 and 0.08, for CIFAR-10, 0.24 and 0.18. The training error, test error, and our bound for the Gibbs posterior average of the 0–1 loss are plotted against $\beta$.

### D.2.4. ULA

We have also conducted experiments using ULA for both datasets. The main difference from SGLD is that we use all the information to compute the gradient at each step. The results are shown in Figure 9.

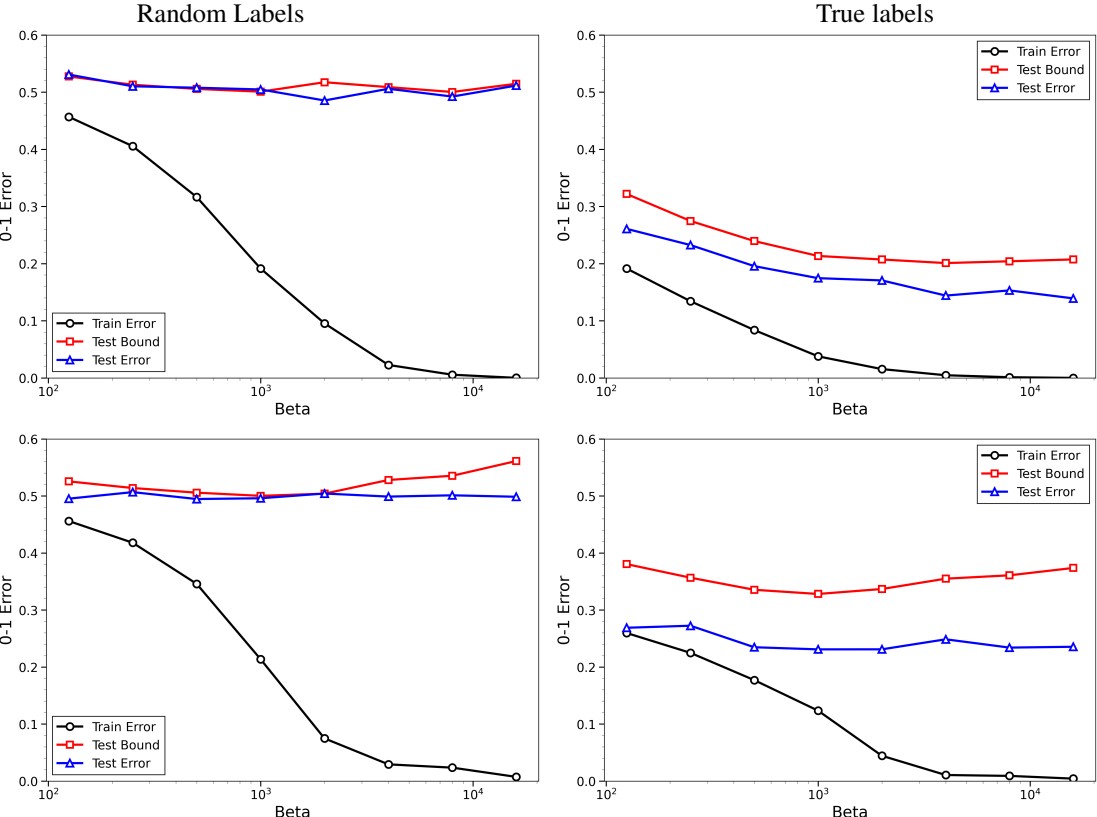

*Figure 9.* ULA on MNIST and CIFAR-10 with 2000 training examples using the BBCE loss function. The first row corresponds to MNIST and the second row to CIFAR-10. Random labels are shown on the left, true labels on the right. Both random and true labels are trained with the same algorithm and parameters on a fully connected ReLU network with one (respectively two) hidden layers of 500 (respectively 1000) units. The calibration factor for MNIST is 0.49, for CIFAR-10, 0.46. Train error, test error, and our bound for the Gibbs posterior average of the 0-1 loss are plotted against $\beta$.

### D.2.5. SAVAGE LOSS FUNCTION

We additionally performed experiments using the Savage loss to verify the robustness of our results across different loss functions. Following the same setup as in the previous section, the outcomes are reported in Figure 10.

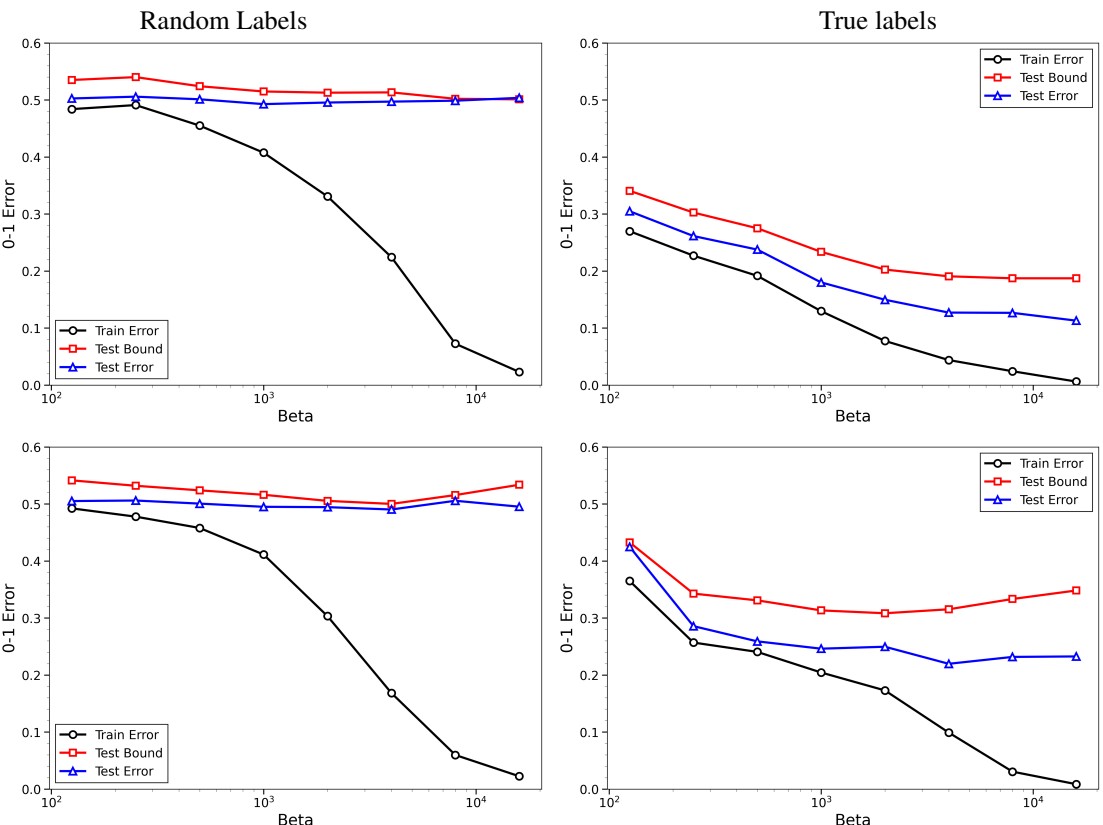

*Figure 10.* ULA on MNIST and CIFAR-10 with 2000 training examples using the Savage loss function. The first row corresponds to MNIST and the second row to CIFAR-10. Random labels are shown on the left, true labels on the right. Both random and true labels are trained with the same algorithm and parameters on a fully connected ReLU network with one (respectively two) hidden layers of 500 (respectively 1000) units. The calibration factor for MNIST is 0.49, for CIFAR-10, 0.59. Train error, test error, and our bound for the Gibbs posterior average of the 0-1 loss are plotted against $\beta$.

### D.2.6. UNBOUNDED LOSS FUNCTION

In this section, we use the binary cross-entropy loss to compute the $\Gamma$ functional. Since binary cross-entropy is unbounded, the loss can become very large at high temperatures. To avoid this issue, we set the standard deviation of the Gaussian prior to 0.1 in this section. The following plot shows the results under the same setup as Section D.2.5, except that we use binary cross-entropy instead of the Savage loss.

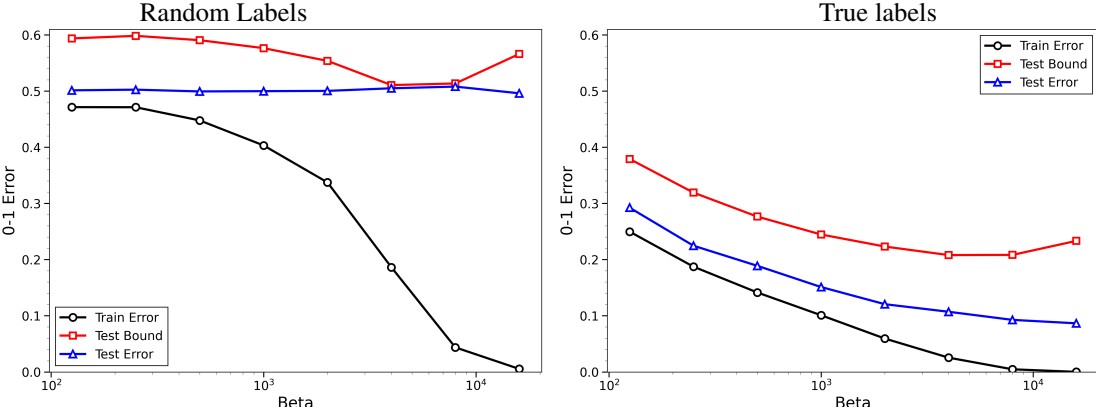

*Figure 11.* ULA on MNIST with 2000 training examples using binary cross-entropy loss function. Random labels are shown on the left, true labels on the right. Both random and true labels are trained with the same algorithm and parameters on a fully connected ReLU network with one hidden layer of 500 units. The calibration factor is 0.34. Train error, test error, and our bound for the Gibbs posterior average of the 0-1 loss are plotted against $\beta$.

### D.2.7. SVHN DATASET

A new dataset has been added to our empirical results in this section, Street View House Numbers (SVHN). Following the exact setting of Figure 4, we trained a 3-layer fully connected ReLU network (1000 units/layer) and a VGG-16 CNN using SGLD on 8,000 SVHN examples. Plotting the training error, test error, and our bound against gives the same predictive behavior observed in Figure 4, as one can see the results in Figures 12, 13. The results confirm the generalizability of our main principle to diverse real-world datasets.

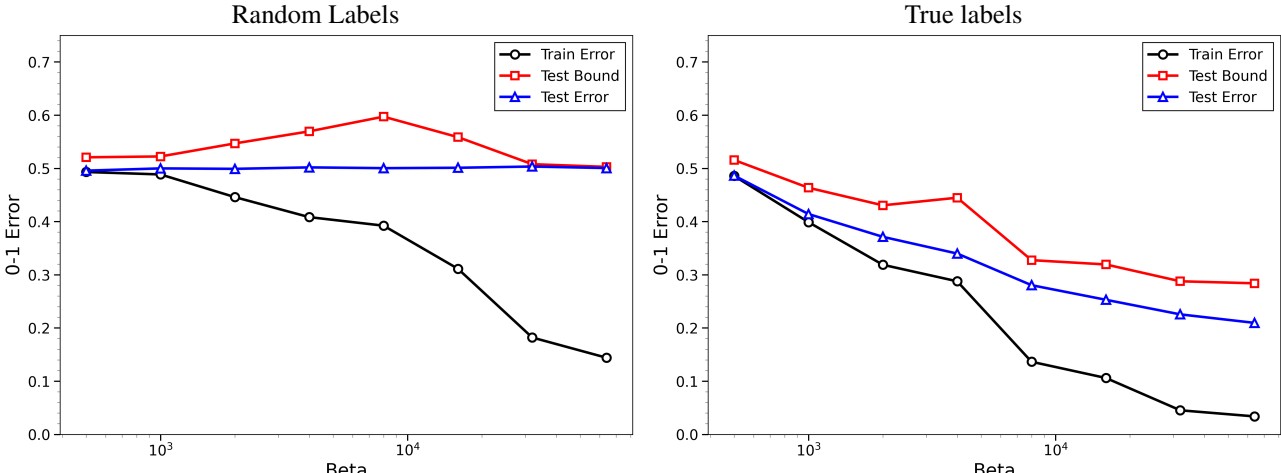

*Figure 12.* SGLD on SVHN with 8000 training examples. Both random and true labels are trained with the same algorithm and parameters on a fully connected ReLU network with three hidden layers of 1000 units. The calibration factor is 0.14. Train error, test error, and our bound for the Gibbs posterior average of the 0-1 loss are plotted against $\beta$.

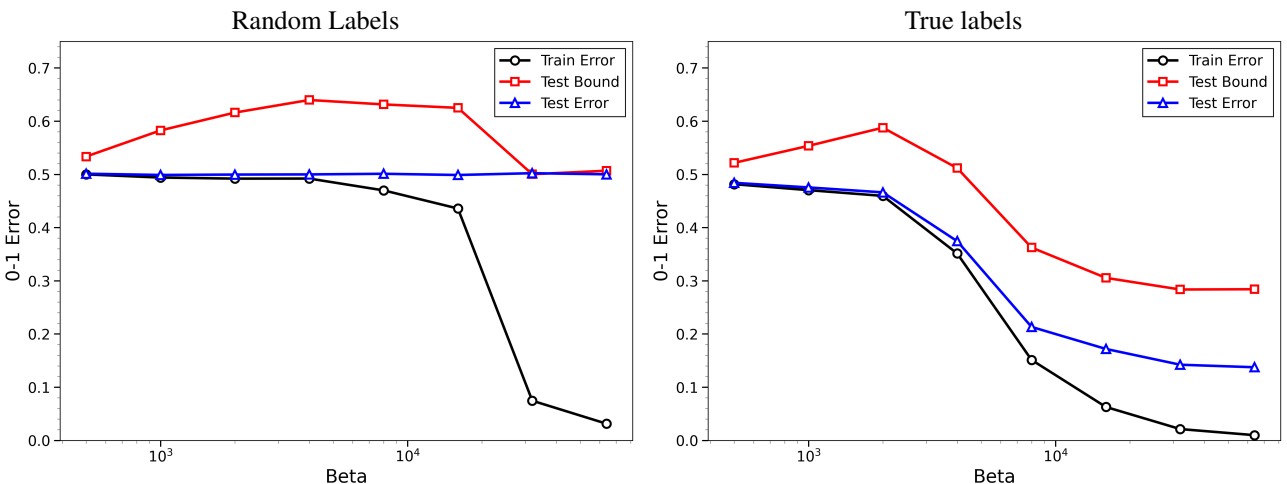

*Figure 13.* SGLD on SVHN with 8000 training examples. Both random and true labels are trained with the same algorithm and parameters on the VGG-16 architecture. The calibration factor is 0.17. Train error, test error, and our bound for the Gibbs posterior average of the 0-1 loss are plotted against $\beta$.

### D.2.8. REAL-WORLD USE CASES

We further evaluated Stochastic Gradient Descent (SGD) to examine the practical relevance of our bounds in real-world interpolation regimes.

Based on our observations, we suggest the following procedure for practitioners who wish to train overparameterized neural networks with standard SGD while also obtaining generalization guarantees. First, randomly permute the labels, train the network at different temperatures, and compute the bound together with the calibration factor. Then, repeat the same procedure using the true labels. At very low temperatures, this approach provides generalization guarantees that may transfer to SGD. The corresponding results are presented in Table 3.

|  | 2HL (W=1000) | 3HL (W=500) | LeNet-5 |
|---|---|---|---|
| Test Error, SGD | 0.0364 | 0.0363 | 0.0308 |
| Test Error, SGLD ($\beta = 64k$) | 0.0498 | 0.0549 | 0.0317 |
| Test Bound, SGLD ($\beta = 64k$) | 0.0860 | 0.1314 | 0.0375 |

*(a)* MNIST, 8k training examples (true labels).

|  | 2HL (W=1500) | 3HL (W=1000) | VGG-16 |
|---|---|---|---|
| Test Error, SGD | 0.1423 | 0.1415 | 0.0933 |
| Test Error, SGLD ($\beta = 64k$) | 0.1719 | 0.1782 | 0.0903 |
| Test Bound, SGLD ($\beta = 64k$) | 0.2266 | 0.2807 | 0.2030 |

*(b)* CIFAR-10, 8k training examples (true labels).

*Table 3.* Comparing SGD test error with SGLD test errors and bounds for different neural network architectures on MNIST and CIFAR-10.

### D.2.9. SENSITIVITY OF THE CALIBRATION SCHEME

Our empirical results are highly robust to the calibration scheme. The calibration factor is derived entirely from random-label data by finding the multiplier that pushes the random-label bound to the known true error ($\frac{1}{2}$ for binary classification). To test stability, we repeated the SGLD experiments as in Figure 4 across 10 different random seeds. The variance of the integrated area is remarkably low: the mean calibration factor for MNIST is $0.75 \pm 0.02$, and for CIFAR-10 it is $0.84 \pm 0.02$. This confirms that the calibration factor is consistently stable and not an artifact of specific random initializations.

