# OpenReview forum: "Generalization of Gibbs and Langevin Monte Carlo Algorithms in the Interpolation Regime"
_ICML.cc/2026/Conference — ICML 2026 regular_

### Official Review · Reviewer_y51w · 2026-03-09

**Soundness:** 2
**Presentation:** 2
**Significance:** 2
**Originality:** 3
**Overall Recommendation:** 4
**Confidence:** 3

**Summary:**

The authors aim to bound the generalization error of Langevin algorithms in the low noise regime via the empirical error in the high noise regime. They plug an integral representation of the logarithm of the density of the Gibbs posterior (Lemma 3.1) into the PAC bayesian theorem, and yield a practical bound by approximating the integral with a finite sum (Theorem 4.5). The difference between this finite sum and the integral form is bounded in Lemma 4.4, under different assumptions.
By utilizing existing results for upper bounds of the KL divergence between discretized algorithms and the Gibbs posterior, they show their bound is applicable to such discretizations under various assumptions, such as the Gibbs posterior satisfying a Log-Sobolev inequality.
Finally, they present experiments showing the tightness of the presented bound for the test error with simple networks under MNIST and CIFAR-10, adjusting their approximation with a calibration factor, calculated by the randomly labeled training data, to tighten the bound in practice.

**Compliance With Llm Reviewing Policy:**

Affirmed.

**Final Justification:**

I thank the authors for their continued and thoughtful engagement with my comments.
I appreciate the clarification they provided regarding prior work, and I agree that the expanded discussion in the final manuscript will improve the paper’s positioning. I am also glad to see that they intend to adopt, at least in large part, my suggestion regarding the organization of Section 4.
I have two final remarks that I hope the authors will take into account in the revision:
1. Regarding the improvement over Harel et al. (2025) mentioned in the rebuttal to reviewer vfJx, it is still not fully clear to me whether this new result is intended to replace the currently presented result, or whether it is meant as an additional alternative result. I suspect this is simply a misunderstanding on my side, and I expect the distinction to become clear in the final manuscript. I also expect the empirical verification to be aligned with it accordingly.
2. On the question of additional empirical baselines, I may not be sufficiently familiar with the literature to suggest the most appropriate comparisons with confidence. One possible reference the authors may wish to consider is Lotfi et al. (2022), though I leave it to the authors to determine whether it is truly comparable in the present setting and whether there are more suitable or more recent works.

Overall, I believe the improvements presented in the rebuttal discussion make the paper a stronger candidate for acceptance, and thus I increased my score to 4.

---
### Reference:
Lotfi, S., Finzi, M., Kapoor, S., Potapczynski, A., Goldblum, M., & Wilson, A. G. (2022). PAC-Bayes compression bounds so tight that they can explain generalization. Advances in Neural Information Processing Systems, 35.

**Key Questions For Authors:**

Nothing to add.

**Limitations:**

Yes (the authors adequately discussed the limitations and potential negative societal impact of their work).

**Strengths And Weaknesses:**

**Strengths**

1. The paper presents a bound on the generalization error for the low temperature regime, where previous bounds, primarily based on PAC-bayesian theory, become vacuous.
2. The underlying idea is compelling and could inspire further research—namely, that “good generalization in the low-temperature regime is already reflected in smaller training errors in the high-temperature regime.” While the converse notion --- that achieving small training error in the high-temperature regime is impossible when generalization is unattainable (e.g., under random labels)---can be inferred from prior work such as Harel et al. (2025) [1], leveraging this intuition to obtain a positive result is a valuable contribution.
3. They show its theoretical applicability to practical discretizations of CLD, illustrating the significance of the work.

**Weaknesses**

1. While the mentioned previous PAC-bayes bounds are uninformative in the low temperature regime, other data-dependent bounds for the generalization error exist, notably Arora et al. (2019) [2] and derivative works, such as the recent Eugenio et al. (2025) [3]. A comparison to these works may better place the paper’s theoretical contribution and significance.
2. More generally, a stronger novelty claim via more rigorous literature review and comparison (maybe in a discussion section) will benefit the paper, especially due to the relative simplicity of the theoretical results presented (Theorem 4.5).
3. The paper is organized in a confusing manner, where the main result, which as I understand is Theorem 4.5, is presented after arguments in favor of its stability under different discretized versions of Langevin dynamics, not necessary for its statement or proof. I would recommend reorganizing the paper for improved clarity.
On the same note, while the comparison to Harel et al. (2025) [1] in lines 260-289 is relevant, I believe it is better moved elsewhere, such as the related literature section or some new discussion section, so as not to disturb the reading flow of the derivation.
4. I find the experimental work lacking in several ways, harming the soundness of the presented work.

  - As the authors admit in lines 378-395, Theorem 4.5 does not in fact yield a practically usable bound, and they instead bound a different quantity presented in Eq. 10.

  - The use of an ad-hoc calibration factor leads the reader to believe the theoretical bound is tighter than it is, seeing that their calibration factor’s values were <1. A discussion of possible causes of this looseness, or an attempt to theoretically justify it, would enhance the paper’s significance greatly. Currently, while as mentioned the resulting empirical bounds are tight, it is hard to regard the empirical work as a demonstration of the theory any more than a heuristic bound, similarly to ones presented in papers mentioned in lines 118-124, while not as usable due to the expensive computational budget required (which I do not believe is a weakness on its own, regarding this as a theoretical paper).

  - Without comparing the empirical results to any baseline, it is hard to establish whether the bound is in fact “tight” compared to similar existing bounds.

I believe many of the weaknesses mentioned may be properly addressed, and I will be happy to raise my scores accordingly.

**Other recommendations**

1. It is unclear how many calculations (e.g., training to convergence under different values of $\beta$) are necessary for the bound to hold. A simple discussion, perhaps by plugging Corollary 4.2 into Lemma 4.4, may shed some light on this.
2. The proof of 4.5, specifically its last part, could benefit from slightly improved clarity.

**Minor issues**

1. Some grammatical/spelling errors  (e.g., “holds”, line 35-right, “an” line 213-right).
2. Low resolution figures 1,3.
3. Fonts in figure 4 should be larger.

**References**

[1] Harel, Itamar, et al. "Temperature is all you need for generalization in langevin dynamics and other markov processes." 39th Annual Conference on Neural Information Processing Systems. 2025.

[2] Arora, Sanjeev, et al. "Fine-grained analysis of optimization and generalization for overparameterized two-layer neural networks." International conference on machine learning. PMLR, 2019.

[3] Clerico, Eugenio, et al. "Generalisation under gradient descent via deterministic PAC-Bayes." 36th International Conference on Algorithmic Learning Theory. 2025.

---

> ### Author Rebuttal · Authors · 2026-03-31
>
> We thank the reviewer for their thorough and constructive feedback. Their comments have directly helped us strengthen our theoretical justifications, particularly regarding the calibration scheme, and improve the paper's overall organization. We are grateful that the reviewer considers increasing their score if they find our answers satisfactory.
>
> ## Weaknesses
>
> **Baselines, Novelty, and Related Work (Arora et al. 2019, Clerico et al. 2025)**
>
> We will add a dedicated discussion section to explicitly contrast our work with existing data-dependent bounds, clarifying our specific contribution. Bounds like Arora et al. (2019) target specific architectures (two-layer ReLU networks) and, similarly to Clerico et al. (2025), track the specific trajectory of gradient descent. In contrast, **our bound applies generally to the Gibbs posterior and is entirely agnostic to the sampling algorithm**. It holds even in discrete settings (e.g., using Metropolis-Hastings) where gradient methods are unavailable.
>
> **Organization and Relation to Harel et al. (2025)**
>
> To improve the structure of the text, we plan to change the current heading of subsection 4.2, where the connection to Harel et al. (2025) is discussed, to "Bounds for Markov Processes." We will include a new theoretical result that directly improves upon Harel et al. (2025). Please see our response to the reviewer vfJx. We will then insert the subsection header "Stability of the Bounds" at line 290, following this discussion.
>
>
> **Theorem 4.5, Equation 10, and Theoretical Justification for Calibration**
>
> While Equation 10 has derived Theorem 4.5 by considering $F = kl$, we put a placeholder, $Q$, which could be either exact as in Theorem 4.5 or approximated by our calibration scheme.
>
>  As the reviewer rightly points out, the strict theoretical bound is loose in practice due to the fundamental difficulty of exact Gibbs sampling in high dimensions, and the bounds in Corollary 4.2 are too coarse to distinguish between different temperatures given realistic step sizes.
>
> To elevate our area-ratio calibration ($A/\bar{A}$) from a heuristic to a **theoretically grounded proxy**, we have found an analytical proof for the strictly convex setting (Bayesian Linear Regression) and possible extension to GLM and NTK regimes. For the details, please see our response to 9HC3.
>
> **Comparison of Empirical Results with Baselines**
>
> To explicitly demonstrate the tightness and practical utility of our approach, we have conducted new baseline comparisons against two distinct categories of prior work (All following material provided in https://anonymous.4open.science/r/ICML-Rebuttal-F42B/Baselines_comparison.pdf):
> * **Vs. Gibbs posterior bounds (Harel et al., 2025)**: explode to vacuous values ($>1$) in our regime of interest. For instance, in Section 4.2, we noted that Harel et al. use the second law of thermodynamics to bound the KL divergence as $KL(\nu_{\beta},\pi) \leq \beta E_{h\sim \pi}[\hat{L}(h)]$. In contrast, we approximate $KL(\nu_{\beta},\pi)$ by $\Gamma(\nu_0^{K-1},x,\beta_0^{K})$, which is smaller even for $\beta < 1$.
> In Figure 1, training standard FCN architectures on MNIST and CIFAR-10, we compare our approximation of the relative entropy ($A$) directly against the relative entropy bound derived from Harel et al. As expected, their bound becomes vacuous for $\beta > n$. Conversely, **our bound on the relative entropy provides sharply tighter, non-vacuous estimates in the low-temperature regime**, even before applying our calibration factor.
> * **Vs. Data-Independent Certificates (Pérez-Ortiz et al., 2021)**: On the other hand, while some works provide non-vacuous bounds for deep networks, they generally rely on data-dependent priors, which is not our focus. Furthermore, as Lotfi et al. (2022) pointed out, these bounds effectively measure validation loss, suggesting that bounds with data-independent priors (like ours) are more informative for fundamentally understanding generalization. Nonetheless, Pérez-Ortiz et al. also offer certificates for data-independent priors. In Table 1, we compare our results on full MNIST against their certificates using the same FCN and CNN architectures with a data-independent prior. Even though their method requires a specialized self-certified training algorithm, **our approach yields tighter certificates for models achieving similar test error**, without needing to alter the standard training objective.
>
> **Other Recommendations and Minor Issues**
>
> **Computational Budget**: We would consider the proposed advice to substitute the estimates of Corollary 4.2 into Lemma 4.4.
>
> **Clarity & Typos**: We will rewrite the final part of the proof for Theorem 4.5 to improve readability, fix all grammatical errors, and increase the resolution/font sizes of Figures 1, 3, and 4.

---

> > ### Author Rebuttal · Reviewer_y51w · 2026-04-04
> >
> > I thank the authors for their thoughtful and constructive rebuttal, and for engaging carefully with my comments.
> >
> > The following concerns are substantially addressed:
> >
> > - Providing a theoretical justification for the calibration argument would strengthen the paper considerably. If incorporated clearly into the revision, this addresses one of my main concerns about the empirical section.
> > - The planned improvements to exposition, proofreading, and figure quality are also welcome.
> >
> > The following points remain open in my view:
> >
> > - I appreciate the authors’ clarification of how their setting differs from prior work. However, I do not believe existing bounds should be set aside solely because they are derived for gradient-based dynamics. The practical relevance of the present paper also rests largely on its applicability to algorithms such as SGLD and related discretizations. For this reason, I still believe a broader discussion of data-dependent bounds that apply in comparable training scenarios is important for properly situating the paper’s significance.
> > - Relatedly, I believe the empirical comparison would benefit from including stronger baselines beyond methods that are already known to become vacuous in the low-noise regime (e.g., Harel et al., 2025). Even if data-dependent approaches ultimately yield tighter bounds, quantifying and discussing that gap would still be informative to the reader and would help clarify the practical contribution of the present work.
> > - Regarding organization, I would still recommend moving the earlier part of Section 4 on discretizations of CLD to after Theorem 4.5, since, as I understand it, these developments are not needed for the statement or proof of the main result. If the authors prefer the current structure, I would appreciate a brief explanation of the rationale.
> >
> > Overall, I appreciate the authors’ detailed response and the proposed revisions, and I would be glad to reconsider my assessment if these remaining issues are addressed clearly in the final version.

---

> > > ### Author Response · Authors · 2026-04-05
> > >
> > > We thank the reviewer for their valuable feedback and for their efforts to improve our manuscript. In the following, we address the remaining concerns.
> > >
> > > 1. **Broader Discussion of Data-Dependent Baselines**‌:
> > > In the final manuscript, we will expand our Related Work section to explicitly contrast our approach with gradient-specific bounds. The bound by Arora et al (2019), which had been suggested for comparison, is specialized to two-layer ReLU networks and derived from special properties of the gradient descent algorithm. It addresses a rather specialized class of models. Clerico et al. (2025), on the other hand, track the evolution of the posterior density along the trajectory of gradient descent and use the single-draw version of PAC-Bayes (part (i) of our Theorem 2.1) to obtain a bound in terms of norms of the Hessian summed over a fixed time horizon. Characteristically, the bound increases with training time, and gives useful results only on a finite-time horizon. This is in common with several results on SGLD or other LMC methods, e.g., Mou et al. (2018), Pensia et al. (2018), cited in our paper. Our improvement of the bound by Harel et al. (2025) (see the response to Reviewer vfJx), however, gives a bound that decreases along the entire training trajectory.
> > >
> > >
> > > 2. **Empirical Comparison**:
> > > We admit that the comparison to Harel et al (2025) was somewhat unfair. We also compared our bound to the certificates of Perez-Ortiz et al. (2021), which are obtained through optimization of a relaxed version of the PAC-Bayes bound, and are data-dependent also in the version without a data-dependent prior (please see again Table 1 at the following link:https://anonymous.4open.science/r/ICML-Rebuttal-F42B/Baselines_comparison.pdf). We welcome other suggestions for data-dependent methods that are appropriate for empirical comparison.
> > >
> > > 3.  **Organization**:
> > > We are inclined to largely follow the reviewer's advice. The message of Theorem 4.5 splits into two parts: a) the convergence to the Gibbs posterior of the algorithm, **for which we want a bound on the generalization error**, and b) the convergence to the Gibbs posterior of the algorithm, **with which we evaluate the bound** (which otherwise remains an unobservable, abstract result). We therefore intend to present a) and b) in this order and, following the reviewer's advice, bring the discussion of the various LMC-methods (with bounds on ULA and CLD) afterwards. We would like to once again thank the reviewer for their suggestions.
> > >
> > > We hope our answers are satisfactory, and we would appreciate final feedback.

---

### Official Review · Reviewer_vfJx · 2026-03-10

**Soundness:** 3
**Presentation:** 2
**Significance:** 3
**Originality:** 2
**Overall Recommendation:** 4
**Confidence:** 1

**Summary:**

The article deals with the problem of approximating the Gibbs posterior in cases where low training errors for data specifically designed to produce very large test errors. High-probability data-dependent bounds are presented both from a hypothesis drawn from the Gibbs posterior and the posterior mean that hold for the entire range of temperatures. The bounds are stable under approximation with Langevin Monte Carlo algorithms.

**Compliance With Llm Reviewing Policy:**

Affirmed.

**Final Justification:**

This is a good contribution, original and technically sound, with some issues in the presentation that can easily be addressed.

The authors have addressed all of my concerns and I have raised my score to 4.

I still need to emphasize that I have very low confidence in my assessment.

**Key Questions For Authors:**

1) The authors use the results of Wisibono and Vempala (2019) to connect with LMC. There are also other results under different assumptions (other than LSI) or other samplers (e.g proximal sampler). Is there a particular reason for the choice of these results?

2) In order to better understand the value of the article's contribution , a comparison with related literature may be needed. Could the authors explain more about the connection with the results of (Harel et al, 2025) ?

**Limitations:**

Yes

**Strengths And Weaknesses:**

Soundness: The results seem sound and well-supported by some proofs. There are also nice experimental results that support the claims.

Presentation: I believe that the presentation is lacking in many different aspects. I think that it is written in a very technical way in the sense that a smoother introduction is needed on the PAC bounds and the problem of interest. A second important drawback is the fact that to my understanding, the contributions are not discussed in a clear and informative way.

Significance: I am not familiar with PAC-Bayesian literature, therefore I don't feel very confident in judging the significance of the results.
I have failed to understand the significance of these results which is probably due to a mix of different factors: my limited understanding of the related literature and the non-smooth presentation of the results.
 I believe that the results are somewhat significant but not up to ICML standards.


Originality: The article introduces some new results that have not appeared in the literature regarding the interpolation regime aiming to bridge a gap between theory and practice. The proof roadmap is not surprising and not particularly difficult.


Overall, I believe that although the article has some value, I believe that it fails short of the ICML acceptance threshold. Given my limited knowledge of the problem, I am very open to change my score if needed.

---

> ### Author Rebuttal · Authors · 2026-03-31
>
> We thank the reviewer for their valuable feedback and address their concerns below. We are grateful that the reviewer considers increasing their score if they find our answers satisfactory.
>
> ## Strengths and Weaknesses
>
>
> **Improving Presentation: An Intuitive Preamble**
>
> To better contextualize our results, the revised manuscript now includes a smoother introduction to PAC-Bayesian Theory:
> * **Standard PAC bounds** are often too pessimistic in the interpolation regime because they account for the worst-case hypothesis in a massive hypothesis space.
> * **PAC-Bayes bounds**, however, measure generalization via the distance (Relative Entropy) between a data-independent prior and a data-dependent posterior.
>
> This framing smoothly transitions to our core principle: the Gibbs posterior—the theoretical minimizer for these bounds—is the ideal lens for understanding why small training errors at high temperatures signal true generalization at low temperatures.
>
> **Clarifying Our Core Contributions and Significance**
>
> We would like to restate a clearer, more intuitive summary of our contributions here.
> * We provide high-probability, data-dependent generalization bounds for the Gibbs posterior that remain non-vacuous across the **entire** temperature range, including the interpolation regime ($\beta > n$). We achieve this via an overlooked integral representation of the log-partition function.
> * **Stability under LMC Approximation**: Our framework bridges the idealized Gibbs posterior and practical algorithms. We prove these bounds are stable under approximations of the Gibbs posterior in relative entropy. By leveraging the second law of thermodynamics for Markov processes, we rigorously extend these bounds to the actual distributions generated by LMC algorithms, such as CLD, HMCMC, and MALA, along their entire training trajectories..
> * **A Practical Calibration Scheme**: We introduce a data-driven calibration step comparing the area under the temperature-loss curve for true vs. random labels. While introduced in the submission as an empirical heuristic, **we have now theoretically justified this calibration step**. As detailed in our response to Reviewer 9HC3, in the setting of Bayesian linear regression (and extending to GLMs and the NTK regime), the label dependency becomes negligible at low temperatures, making our assumed ratio exact. This theoretically backed step yields **remarkably tight upper bounds** on test error for overparameterized networks. We have added these formal derivations to the Appendix. (Please also see our response to Reviewer y51w for empirical comparisons demonstrating this tightness).
>
> **Proof Complexity**
>
> While the mathematical identities we use, such as Lemma 3.1, are relatively simple and rooted in statistical physics, **we view this mathematical elegance as a strength**. Our novelty and main contribution, therefore, lie in how we combine these results to explain the puzzling phenomenon of generalization in overparameterized neural networks.
>
>
> ## Key Questions
>
> **Choice of Sampler**
>
> We used Vempala & Wibisono (2019) because their results guarantee convergence in relative entropy, which our stability result requires. However, **our framework is entirely modular**. It is not restricted to ULA or SGLD. If a practitioner uses a more advanced sampler (like a proximal sampler) with a tighter relative entropy bound, that bound plugs directly into our Theorem 4.3 to get an even tighter generalization guarantee.
>
>
>
> **Connection to Harel et al. (2025)**
>
> Combining the methods of Harel et al with the integral representation of the log-partition function leads to the following bound for Markov processes initialized from the prior:
> $$KL\left( \nu_{\beta,t},\pi \right) \leq \lambda_{t}\beta E_{h\sim\pi}[\hat{L}(h)] +\left( 1-\lambda_{t}\right) \int_{0}^{\beta}E_{g\sim G_{\gamma }}\left[ \hat{L}\left( g\right) \right] d\gamma$$
>
> where $\lambda_{t}=KL\left( \nu_{\beta,t}, G_{\beta }\right) /KL\left( \pi, G_{\beta }\right)$.
> If $G_{\beta }$ is invariant, **this strictly improves upon the bound of Harel et al. (2025)** for all $t>0$ on the entire training trajectory. If $\nu_{\beta,t}\rightarrow G_{\beta }$, it converges to a bound for the Gibbs posterior with direct implications for the interpolation regime. This applies to CLD, MALA, and HMCMC. We have added this explicit comparison to the revised paper.

---

> > ### Author Rebuttal · Reviewer_vfJx · 2026-04-01
> >
> > I thank the authors for addressing my concerns. I will raise my score.

---

> > > ### Author Response · Authors · 2026-04-03
> > >
> > > We thank the reviewer for considering our rebuttal and raising their score.

---

### Official Review · Reviewer_9HC3 · 2026-03-10

**Soundness:** 3
**Presentation:** 3
**Significance:** 3
**Originality:** 3
**Overall Recommendation:** 5
**Confidence:** 4

**Summary:**

The paper addresses a key question in theoretical machine learning. A sufficiently overparameterised machine learning model can achieve very low training loss on both real, meaningful data as well as on impossible data, such as random labels. In the latter case however, the model fails to generalise to test data. Observing near-zero training error therefore, does not necessarily mean that the model has learned to extract meaningful features from the data, hence training loss along with the hypothesis space is not a good predictor of generalisation behaviour. The authors study the Gibbs posterior - a distribution over model parameters that gives more weight to parameters with lower training loss - and connect it to practical sampling methods like SGLD and ULA. The paper argues that, in order to predict generalisation behaviour, one should look at how the  mean training loss under the Gibbs posterior changes as the inverse temperature  varies. The authors derive a theoretical generalisation bound, which  depends on the whole path of mean training loss of the Gibbs posterior across different temperatures, not just the final training loss. They also empirically demonstrate on the MNIST and CIFAR-10 benchmarks, that the mean training loss on real-label data is lower much earlier along the temperature path than random-label data.

**Compliance With Llm Reviewing Policy:**

Affirmed.

**Final Justification:**

It seems a strong paper to me and the review rebuttal has improved my confidence.

**Key Questions For Authors:**

1. Could the calibration step used in the experimental section be justified theoretically, perhaps in some simple setting?
2. How sensitive are the empirical results to the calibration scheme?
3. Do the authors expect the same behavior to persist on larger-scale datasets or modern training pipelines beyond the current setups?

**Limitations:**

The theoretical idea is very interesting but the empirical results depend on a calibration scheme that is not theoretically justified.

**Strengths And Weaknesses:**

Strengths
• The paper addresses an important theoretical question: in the interpolation regime, near-zero training error can occur both on meaningful data and on random-label data, so training error alone no longer explains generalization. • The main technical idea is very interesting. The KL term for the Gibbs posterior is written in terms of an integral over the mean training loss as the inverse temperature varies. This gives an explanation as to why behaviour/mean training loss at smaller ’s can already act as a predictor of generalisation performance in the low-temperature regime. • The paper first derives bounds for the exact Gibbs posterior, and then studies how these bounds change when the Gibbs distribution is approximated by Langevin-based samplers, such as SGLD and ULA. • The experiments confirm the paper’s theoretical predictions. On MNIST and CIFAR-10, the experiments compare a true-label and random-label dataset across different temperatures and it is very interesting to see that the theoretical test error bounds are indeed informative and often quite tight.

Weaknesses:
• One reservation is that the practically strongest bounds rely on a calibration step that seems to be empirically effective, but is not rigorously justified by the theory. • The empirical validation results seem very promising but are mainly centered on MNIST, and CIFAR-10. • In figure 4, it would be visually helpful to have a title for each column (random and correct labels respectively). A title for each row (MNIST, CIFAR-10) might also be helpful.

Specific justification for soundness score: The theoretical results seem meaningful and technically interesting although I did not thoroughly go through the proofs.

Specific justification for Presentation Score:   Readable and well presented.

Specific justification for Significance score: The problem is important, and the paper offers a genuinely useful perspective on generalization in the interpolation regime.

Originality:  The path-based view of the Gibbs posterior complexity term via mean training loss across different temperatures seems to be a novel and very interesting approach.

---

> ### Author Rebuttal · Authors · 2026-03-31
>
> We thank the reviewer for their positive assessment and constructive feedback. We have addressed the core questions regarding the calibration scheme's theoretical justification and empirical robustness, and expanded our experimental scope.
>
> ## Strengths and Weaknesses
>
>
> **New Dataset: SVHN**
>
>
> To strengthen our empirical validation beyond MNIST and CIFAR-10, we conducted new experiments on the **Street View House Numbers (SVHN)** dataset. Following the exact setting of Figure 4, we trained a 3-layer fully connected ReLU network (1000 units/layer) and a VGG-16 CNN using SGLD on 8,000 SVHN examples. Plotting the training error, test error, and our bound against $\beta$ gives the same predictive behavior observed in Figure 4 (see Figs. 1 and 2 in the link [1]). The results confirm the generalizability of our main principle to diverse real-world datasets.
> [1]: https://anonymous.4open.science/r/ICML-Rebuttal-F42B/SVHN_dataset.pdf
>
> **Figure 4 Clarity**
>
> As suggested, we will update Figure 4 (and related figures) to include explicit row titles (MNIST, CIFAR-10) and column titles (Random Labels, Correct Labels) to improve visual readability.
>
> ## Key Questions
>
> **Theoretical Justification of Calibration (Q1)**
>
> While the calibration step was initially introduced as an empirical heuristic in our submission, the reviewer’s question prompted us to investigate a formal grounding. We are pleased to report that we can now theoretically justify the scheme in the setting of **Bayesian linear regression**. Assuming a quadratic loss function with an isotropic Gaussian prior, the Gibbs posterior is Gaussian. We show that the stationary distribution of the Unadjusted Langevin Algorithm (ULA) is also Gaussian with the same mean but inflated variance. In this exact setting, both the KL divergence (between ULA's invariant distribution and the prior) and the integral term derived from Lemma 3.1 can be computed analytically. Crucially, the label dependency appears only in a constant term, which becomes negligible in the low-temperature regime. Therefore, our assumed ratio is theoretically correct. Furthermore, assuming the ULA approximation bound by Vempala & Wibisono (2019) is sharp, similar reasoning demonstrates this ratio is fixed for **Generalized Linear Models (GLMs)** and the **Neural Tangent Kernel (NTK)** regime. We will include these formal derivations in the Appendix of the next version.
>
> **Sensitivity of the Calibration Scheme (Q2)**
>
> Our empirical results are highly robust to the calibration scheme. The calibration factor $r$ is derived entirely from random-label data by finding the multiplier that pushes the random-label bound to the known true error ($0.5$ for binary classification). To test stability, we repeated the Figure 4 SGLD experiments across **10 different random seeds**. The variance of the integrated area $\bar{A}$ is remarkably low: the mean calibration factor $r$ for MNIST is **0.75** **($\pm$ 0.02)**, and for CIFAR-10 it is **0.84** **($\pm$ 0.02)**. This confirms that the calibration factor is consistently stable and not an artifact of specific random initializations.
>
>
> **Scalability to Modern Training Pipelines (Q3)**
>
> We firmly expect our core principle—that good generalization at low temperatures is signaled by rapid loss minimization at high temperatures—to hold across larger scales and modern architectures (e.g., Transformers). This is a fundamental property of the Gibbs distribution. While explicitly simulating MCMC chains to full convergence for ImageNet-scale data remains a well-known computational bottleneck for the broader community, our theoretical framework naturally scales to any setting where the Gibbs posterior can be adequately approximated.

---

> > ### Author Rebuttal · Reviewer_9HC3 · 2026-04-03
> >
> > This was already a very strong paper and I think it has been improved.

---

> > > ### Author Response · Authors · 2026-04-05
> > >
> > > We thank the reviewer again for their positive evaluation and for recognizing our paper as significant.

---

### Official Review · Reviewer_EiRp · 2026-03-12

**Soundness:** 3
**Presentation:** 3
**Significance:** 3
**Originality:** 3
**Overall Recommendation:** 5
**Confidence:** 3

**Summary:**

The paper first pointed out the fact that the generalization error for the Gibbs-posterior at inverse temperature $\beta$ depends on the integrated training error of the Gibbs-posterior path from 0 to $\beta$. This gives the insight that higher training error at high temperatures may indicate high test error. If this speculation proves true, then we can finally explain the phenomenon of low training error for impossible labels in the overparameterized regime. The authors then provided a way to estimate the generalization error for Gibbs-posteriors along the interpolation path using LMC such that the estimates can be compute solely from training error. Finally, they show empirically that their generalization bound estimate is tight compared to the test error.

**Compliance With Llm Reviewing Policy:**

Affirmed.

**Key Questions For Authors:**

Here are my questions
1. Since $\nu_0^K$ approximates an interpolation path, why not use a more specialized algorithm in the annealing literature such as parallel tempering?
2. What is the rational for choosing the temperature grid in the experiments and how does it affect the bound?

**Limitations:**

yes

**Strengths And Weaknesses:**

**Soundness**: My concern is that LMC is a local MCMC and may fail to approximate multi modal distributions, especially in low temperatures where distribution modes are well separated. Also, it is a pity that the authors could not decisively determine the relationship between training error at high temperatures and test error.

**Presentation**: I believe that this work is well and clearly written for the most part.

**Significance**: The insight from this paper is quite valuable in my opinion as it presents a plausible explanation to the low training error for impossible labels in the overparameterized regime phenomenon. If one could, for example, construct a lower generalization bound with a similar term then the explanation would be complete.

**Originality**: The bound constructed in the paper made use of interpolation path for the Gibb-posterior which has connection to annealing literature. As far as I know, no work in PAC bounds has utilized this approach.

---

> ### Author Rebuttal · Authors · 2026-03-31
>
> We thank the reviewer for recognizing our paper's originality, and its significance in explaining overparameterized learning phenomena. We address your specific questions below and will incorporate these clarifications into the final version.
>
> ## Strengths and Weaknesses
>
> **Lower Bounds and High-Temperature Dynamics**
>
> We agree with your insightful suggestion regarding a lower bound. A lower bound with a similar term would definitely complete the explanation, allowing us to **decisively determine the relationship between training error at high temperatures and test error**. We will explicitly discuss this as a primary open problem for future work in the final version of the paper.
>
> ## Key Questions
>
> **Choice of MCMC Algorithm (LMC vs. Parallel Tempering)**
>
> While advanced annealing methods like parallel tempering are theoretically appealing for navigating well-separated modes, their **computational and memory overheads are strictly prohibitive for modern, overparameterized deep neural networks**. We specifically chose LMC (via SGLD/ULA) because it **aligns with the standard optimizers (SGD) utilized by practitioners**. We agree that sampling from multimodal distributions at low temperature is challenging. However, the solutions found by SGLD/ULA show great generalization in practice. By using SGLD, we demonstrate that **our bounds are directly applicable to the algorithms practitioners actually deploy**, and support our core principle, that good generalization at low temperatures is signaled by rapid loss minimization at high temperatures.
>
> **Rationale and Effect of the Temperature Grid**
>
> We utilized an exponentially spaced temperature grid to efficiently capture the critical dynamics of the interpolation path. The integral requires dense estimation in the **rapid-decay (high-temperature)** regime, where the training error drops abruptly, while still needing to reach very low-temperatures.
>
> _Effect on the bound_: A finer temperature grid gives a more precise numerical approximation of the integral, which **directly tightens the bound by reducing overestimation errors**. However, this comes at a linear increase in computational cost. Our geometric grid achieves a **pragmatic trade-off**, ensuring the computed bound remains tight without incurring unmanageable compute times.

---

> > ### Author Rebuttal · Reviewer_EiRp · 2026-04-03
> >
> > I thank the authors for their response. Most of my questions and concerns have been addressed so I will keep my score as is.
> >
> > There is one comment I would like to add about parallel tempering (PT). Note that PT is MCMC step + communication step.
> > In the paper, LMC is already used to approximate each $G_{\beta_k}$ so applying PT on top is simply adding a communication step between each LMC step. The benefit is that you can do less iterations for colder chains as samples from warmer ones improve their mixing time. In short, PT will only marginally increase your memory (an additional $K$-length vector to keep track of the temperatures) and only marginally increase computation per step (communication step is $O(1)$ compared to $O(dn)$ MCMC step for each chain). See [1] for more details. Ultimately, this is an implementation choice and does not affect the overall message of the paper so I won't pursue any further.
> >
> > [1] Syed S., Bouchard-Côté A., Deligiannidis G., Doucet A. (2022) Non-reversible parallel tempering: A scalable highly parallel MCMC scheme. Journal of the Royal Statistical Society: Series B (Statistical Methodology), 84, 321–350.

---

> > > ### Author Response · Authors · 2026-04-05
> > >
> > > We thank the reviewer again for their positive assessment and plan to study and possibly incorporate the interesting reference in our manuscript.

---

### Decision · Program_Chairs · 2026-04-30

**Decision:**

Accept (regular)

**Comment:**

This paper analyzes generalization performance of DNN and relate it to training performance of samples from the Gibbs-posterior with higher temperatures.  Specifically, the generalization error is theoretically bounded by the are under the curve of training error over temperatures.  The theoretical results provide insight into the phenomenon of low training error on random labels in the overparameterized regime, in a manner similar to Harel et al. (2025).  The authors additionally propose to practically estimate the generalization error by performing Langevin Monte Carlo sampling, and empirically show that the estimated generalization bound is not loose.

All reviewers acknowledged significant contributions and impact into the ML community.  Minor concerns raised have been well addressed.